# Dynamic Modeling of Carbon Dioxide Transport through the Skin Using a Capnometry Wristband

**DOI:** 10.3390/s23136096

**Published:** 2023-07-02

**Authors:** Pierre Grangeat, Maria-Paula Duval Comsa, Anne Koenig, Ronald Phlypo

**Affiliations:** 1CEA, Leti, MINATEC Campus, Université Grenoble Alpes, F-38000 Grenoble, France; pierre.grangeat@wanadoo.fr (P.G.); mariapaula.comsa@gmail.com (M.-P.D.C.); 2CNRS, Grenoble INP, GIPSA-Lab, Université Grenoble Alpes, F-38000 Grenoble, France; ronald.phlypo@grenoble-inp.fr

**Keywords:** capnometry, wearable health device, continuous monitoring, blood carbon dioxide concentration, carbon dioxide transcutaneous pressure, convection diffusion equation, compartmental model, state-space Markovian model, augmented dynamical system, Kalman filter

## Abstract

The development of a capnometry wristband is of great interest for monitoring patients at home. We consider a new architecture in which a non-dispersive infrared (NDIR) optical measurement is located close to the skin surface and is combined with an open chamber principle with a continuous circulation of air flow in the collection cell. We propose a model for the temporal dynamics of the carbon dioxide exchange between the blood and the gas channel inside the device. The transport of carbon dioxide is modeled by convection–diffusion equations. We consider four compartments: blood, skin, the measurement cell and the collection cell. We introduce the state-space equations and the associated transition matrix associated with a Markovian model. We define an augmented system by combining a first-order autoregressive model describing the supply of carbon dioxide concentration in the blood compartment and its inertial resistance to change. We propose to use a Kalman filter to estimate the carbon dioxide concentration in the blood vessels recursively over time and thus monitor arterial carbon dioxide blood pressure in real time. Four performance factors with respect to the dynamic quantification of the *CO*_2_ blood concentration are considered, and a simulation is carried out based on data from a previous clinical study. These demonstrate the feasibility of such a technological concept.

## 1. Introduction

Respiration is a basic metabolic function of living organisms. On the one hand, it includes oxygen intake, which is used to produce energy at the cellular level. On the other hand, the $CO_{2}$ that is freed during the energy production is evacuated back into the environment. Most gas transportation within the organism is achieved via the circulatory system, whereas the gas exchange between the organism and the environment occurs at the level of the alveolocapillary barrier, although a secondary part of respiration takes place through the skin barrier [1].

The $CO_{2}$ in the blood is dissolved partly in its molecular species (aqueous carbon dioxide $CO_{2\;\left(aq\right)}$ and carbonic acid $H_{2}CO_{3}$) and partly—due to the alkaline property of blood—in its deprotonated, ionic species (as carbonate $CO_{3}^{2-}$ or bicarbonate, which is also called hydrogen carbonate, $HCO_{3}^{-}$). The proportion of $CO_{2}$ dissolved in plasma determines the pH of the blood [2]. The carbon dioxide pressure $\left(P_{CO_{2}}\right)$ is linked to the concentration of the molecular species $(C_{CO_{2}}\overset{\mathrm{not}}{=}\;\left[CO_{2}\right])\overset{\mathrm{not}}{=}$ for “notation”. Thus, an increase in the concentration of dissolved carbon dioxide induces a decrease in the blood pH. Metabolism is optimal for a pH of the blood in the range of 7.35 and 7.45, i.e., slightly alkaline [3]. It is the renal system, and the carbonic anhydrase enzyme in particular, that controls the blood’s pH in the human body. An excess in pH might induce alkalosis and a deficit an acidosis. Both have a severe impact on human metabolism and the proper functioning of human organs [4]. Consequently, the monitoring of $P_{CO_{2}}$ by capnometry is of major clinical interest.

Indeed, an increase in the carbon dioxide concentration might be a consequence of, for instance, the respiration track that is injured or obstructed, no longer evacuating $CO_{2}$. This may not only happen in the case of chronic obstructive pulmonary disease (COPD) but also when an infectious disease, such as COVID-19, affects the lung. Capnometry is also useful to detect alveolar hypoventilation or hypercapnia. For clinical use, monitoring the carbon dioxide level is important for the follow-up of anesthesia or mechanically assisted respiration. It is used, among other things, for the monitoring of newborns in incubators. The main method to monitor respiration is to control the composition of the inhaled and exhaled air. An alternative method is to study gas exchange through the skin. When considering the development of a wearable device, this alternative solution is attractive. Controlling the air at the mouth or nose implies wearing a mask or a cannula, which is uncomfortable for longitudinal studies. Monitoring skin respiration—for instance, at the forearm—is more comfortable for the user. Developing a capnometry wristband would be of high benefit for homecare patient monitoring. However, it is a technological challenge to measure and monitor the low gas concentrations in a robust and reliable way. To this end, accurate instrumentation is necessary, in addition to relevant modeling and data processing. The model contributes to optimizing the design of the sensing device, by constructing the direct path on which the data processing will rely, and generating synthetic data to test the signal processing.

Several review papers on capnometry have been published [5,6,7,8,9,10]. The standard technology to measure carbon dioxide transcutaneous pressure is based on Severinghaus’s electrochemical principle [11]. To obtain a shorter response time, we opt here for the optical non-dispersive infrared (NDIR) principle. This technology has been first introduced in the pioneering works of Thiele and Eletr et al. [12,13]. Some recent papers are proposing to use optical sensors based on fluorescent or luminescent dyes [10,14,15]. Currently, most of the capnometry devices rely on a model for a steady-state, static equilibrium between the carbon dioxide pressure within the blood and within the measurement device. In addition, most of the devices are based on a closed chamber measurement principle. In this work, we propose to consider an open chamber principle. Technology that combines, sequentially, an accumulation of carbon dioxide in a closed loop and a circulation of a nitrogen flow to flush the accumulated carbon dioxide has also been proposed [14,16,17,18]. In the article of Iitani et al. [18], the accumulation phase lasted 60 s and the flushing lasted 30 s. The measurement is carried out at the beginning of the accumulation phase. The concentration is computed from the measurement of the accumulation flow rate. Each measurement is separated by the duration of the entire measurement cycle, which is one minute and thirty seconds.

In this work, we suggest using an open chamber principle including a continuous air flow. The use of nitrogen as carrier gas is not convenient for an autonomous wearable device. Hence, we propose to introduce air convection in the collection cell to speed up the carbon dioxide diffusion through the skin into the ambient air. However, this also comes with a dilution of the $CO_{2}$ concentration in the measurement cell. Hence, we need to study the compromise between the response time and the carbon dioxide gas concentration within the measurement cell. In order to study such an open chamber principle, we propose a model for the temporal dynamics of the carbon dioxide exchange between blood and the gas channel within the measurement device.

Signal processing methods include a large variety of approaches. An introduction is presented in the book chapter by Grangeat on signal processing from *Nanoscience: Nanobiotechnology and Nanobiology* [19], with a section dedicated to data extraction, including sub-sections on extracting physical quantities, systems approach, inverse problems, and regularized solutions, and sections on sensor correction and data analysis. In a paper by Marco and Gutiérrez-Gálvez on signal and data processing for machine olfaction and chemical sensing [20], they explain that relevant signal processing is mandatory to improve the robustness of the instrumentation. They establish a clear distinction between regression techniques to improve quantification, classification for the identification of the species, and clustering for grouping species of similar properties. Our objective is to develop a model-based signal processing approach for regression as described in a book by Candy [21]. We propose a model based on physical equations, describing the main physical and physiological parameters that must be taken into account for estimating the carbon dioxide concentration in the blood. This estimated value can then be incorporated in classification techniques associated with statistical models, combined with feature selection techniques, for patient diagnosis purposes, as described, for instance, in the article by Al Fahoum et al. [22] for the identification of coronary artery diseases using photoplethysmography signals. Here, we introduce a dynamic model for the dynamic monitoring of carbon dioxide blood content. We also propose to use a Kalman filter [23,24,25,26] to estimate the carbon dioxide concentration in the blood vessels recursively over time and thus to monitor the carbon dioxide blood pressure in real time to create a signal processing method that can be embedded in the wristband. Whereas static equilibrium is associated with Fick’s first-order diffusion equation, any dynamic model must be based on Fick’s second-order diffusion equation. Hence, carbon dioxide transportation can and will be modeled by convection–diffusion transport equations.

The exchange of carbon dioxide, inert gas, and volatile organic compounds between blood and air has been studied in detail to describe lung respiration [27,28,29,30,31]. Skin respiration has been reported for amphibians [32] but is less explored in humans [33,34]. Due to the effect of $CO_{2}$ on the atmosphere and its impact on climate, $CO_{2}$ emission has been studied in many application fields related to environmental studies. For instance, the emission of $CO_{2}$ is associated with soil respiration, water respiration, vegetation respiration, and fruit respiration. Models associated with this type of respiration have also been reported. The exchange of carbon dioxide between a moving gas phase and a moving liquid phase through a membrane has been studied extensively for a hollow-fiber membrane extractor [35,36,37,38,39]. The objective here is to capture carbon dioxide gas from the environment to fix it in the liquid phase. The $CO_{2}$ is transported in the opposite direction with respect to our desorption case, i.e., when studying diffusion through the skin barrier from the liquid (blood) phase to the gas (air) phase.

Although closed chambers are suitable for studying the emission flow rate of the carbon dioxide pressure at a static equilibrium [40], in order to study the temporal evolution of the $CO_{2}$ blood concentration, we propose an open chamber sensor architecture. The $CO_{2}$ will be evacuated using air flow through the collection cell. The air flow will be produced by an external pump. To keep the time delay between carbon dioxide emission and carbon dioxide measurement as short as possible, the NDIR optical measurement will be located as close as possible to the surface of the skin. A schematic description of the measurement principle is depicted in Figure 1. Our model is divided in four compartments: blood, in which the unknown concentration is to be estimated; skin, which behaves like a membrane; the measurement cell, in which the NDIR measurement is actually carried out; and the collection cell, where air convection is forced.

From a technological point of view, this new device is based on a previous version of the CAPNO device (a capnometry wristband device named CAPNO), as described in our previous articles [41,42]. The main difference here is that the measurement cell is placed directly at the contact of the skin; the flow through the measurement cell is transverse, and it relies only on diffusion. Additionally, the flow convection that was induced only by the gradient of temperature in the thermofluidic channel is now induced by an air flow pump placed outside the collection cell, either at the input or at the output. However, the skin is still heated by a thermocircuit placed over the grid in contact with the skin, and the measurement is achieved by a near-infrared (IR) measurement combining an IR black body source placed on one side of the measurement cell and two thermopiles on the opposite side. The IR radiation propagates along the measurement cell transversally with respect to the propagation direction of the carbon dioxide, which is normal to the skin surface, including multiple reflections on the lower grid in contact with the skin and the upper grid in contact with the collection cell. Two filters in front of each thermopile select the light wavelength to achieve the dual-wavelength measurement. We use a detection wavelength of 4.26 µm corresponding to a peak of the absorption intensity of carbon dioxide and a reference wavelength of 3.91 µm corresponding to a minimum of the absorption intensity for water. To obtain an autonomous wearable device, the electronics are embedded. They drive both the data acquisition, the pump, the heating, and the data exchange using Bluetooth Low Energy (BLE) protocols. Microelectromechanical system sensors (MEMS) are placed in each cell to measure continuously the temperature, pressure and relative humidity.

The objective of this article is to describe a dynamic model of carbon dioxide transport through the skin on a capnometry wristband (named CAPNO) in order to introduce the theoretical framework for the development of this wearable device and to prove with simulated data the feasibility of this open chamber technology concept for the continuous monitoring of carbon dioxide blood concentration [43]. In Section 2, on Materials and Methods, we present the dynamic model that describes the $CO_{2}$ transport from blood-irrigated tissues through the skin to the measurement and the collection cells, followed by the methods for model evaluation. In Section 2.1, we present the continuous space and time dynamical model, including, in Section 2.1.1, a description of the four compartments’ geometry (blood, skin, measurement and collection cells); in Section 2.1.2, a description of the measurement of the $CO_{2}$ concentration with dual-wavelength IR thermopiles; in Section 2.1.3, the assumptions leading to the boundary conditions; and, in Section 2.1.4, the physiological and physical parameters on which the model relies. Then, in Section 2.2, we describe the discrete state-space representation of $CO_{2}$ transport for the development of the simulation and inference software. This defines the direct model with the spatial discretization described in Section 2.2.1, the temporal discretization described in Section 2.2.2, the exogenous inputs described in Section 2.2.3, and the state-space model and the associated noise model described in Section 2.2.4. In Section 2.3, we present the model inversion to estimate the carbon dioxide concentration $C_{CO_{2}}^{in\;blood}$ and the associated pressure $P_{CO_{2}}^{in\;blood}$ at the input of the blood compartment from the carbon dioxide concentration $C_{CO_{2}}^{meas}$ computed from the NDIR optical measurement in the measurement cell. In Section 2.3.1, we derive a Kalman filter for the recursive estimation of the blood carbon dioxide concentration. In Section 2.3.2, we derive from the carbon dioxide blood concentration the carbon dioxide blood pressure. Then, in Section 2.4, we introduce the four performance parameters that we propose to characterize the quantification of the $CO_{2}$ blood concentration and its dynamic behavior. In Section 3, we present the results on simulated data, the performance parameters for the direct problem, the inverse problem and the complete model. We show also the performance parameters on a simulation of a realistic clinical test. In Section 4, we propose a discussion on these results and on the potential applications of this dynamic model. In Section 5, we present a general conclusion on the feasibility of this innovative capnometry wristband concept. We also discuss the contribution of the model to designing accurate instrumentation, to computing simulated data and to developing model-based signal processing.

## 2. Materials and Methods

The model is built upon the four compartments described in Figure 1. We consider a simplified one-dimensional (1D) diffusion–convection model of the transport equation along the z-axis perpendicular to the skin surface.

### 2.1. Continuous Space–Time Model of the $CO_{2}$ Transport

To study the transport of $CO_{2}$ species through the different media, we will start by writing down the conservation laws. Given a number N of moles of the species in a compartment of limited volume V, consider the net flux J (in mol·s^−1^) through its boundary Ω (according to the surface normals pointing outward from the volume), as illustrated in Figure 2.

The transport equation is based on the conservation law of species in fluid mechanics [44]. The number of moles of the species in the volume changes with time according to the net flux J through the surface. Hence, writing down the number of moles within the volume V as a function of time leads to the following equation:(1)$$N\left(t+\delta t\right)=N\left(t\right)+\int_{t}^{\delta t}-J\left(\tau \right)\;d\tau$$
where the minus sign stems from the fact that the surface normals are pointing outwards. If we consider an infinitesimal time step, we find that
(2)$$\dot{N}\left(t\right)=\underset{\delta t\to 0^{+}}{\lim}\frac{N\left(t+\delta t\right)-N\left(t\right)}{\delta t}=\underset{\delta t\to 0^{+}}{\lim}\frac{1}{\delta t}\int_{t}^{\delta t}-J\left(\tau \right)\;d\tau =-J\left(t\right)$$

The net flux through the surface Ω that governs the dynamics of the quantity N is an integration of local fluxes j (in mol·m^−2^·s^−1^) flowing through the boundary according to the surface normal $\vec{n}\left(r\right)$ for all r ∈ Ω. We may thus write
(3)$$\dot{N}\left(t\right)=-J\left(t\right)=\int_{\Omega }-j\left(r,t\right)\cdot \vec{n}\left(r\right)\;dr$$

The fluxes j(r,t) can be decomposed as a sum of three contributions:

A diffusive flux $j_{\mathrm{diff}}\left(r,t\right)$, which is given by Fick’s first law of diffusion [45,46,47,48]:

(4)$$j_{diff}\left(r,t\right)=-D\left(r\right)\nabla C\left(r,t\right)=D\left(r\right)\nabla \frac{N\left(r,t\right)}{dV\left(r\right)}=D\left(r\right)\frac{\nabla N\left(r,t\right)}{dV\left(r\right)}$$
where the latter equality holds true only if the infinitesimal compartments are of constant volume (which is the case in Euclidean coordinates but is not the case in spherical nor in cylindrical coordinates). The diffusion constant D(r) depends on the medium and the species of $CO_{2}$.

A convective flux $j_{\mathrm{conv}}\left(r,t\right),$ which is the media’s convective flow, transporting the species with a speed u(r) in m·s^−1^:


(5)
$$j_{\mathrm{conv}}\left(r,t\right)=u\left(r\right)C\left(r,t\right)$$


Noting that a change in moles for a specific volume can now be expressed with the aid of the divergence theorem as
(6)$${\dot{N}}_{V}\left(t\right)=\frac{\partial }{\partial t}\int_{V}C\left(r,t\right)=\int_{\Omega }-j\left(r,t\right)\cdot \vec{n}\left(r\right)dr=\int_{V}\nabla \left(D\left(r\right)\nabla C\left(r,t\right)-u\left(r\right)C\left(r,t\right)\right)dr$$

This is a convection–diffusion equation that is the equivalent of Fick’s second law [45,46,49] of a species’ concentration within a transporting medium that allows for a continuous concentration profile that is differentiable almost everywhere. In this case, we find the dynamics’ equations by considering that the left- and right-hand side must be valid for any subvolume; hence, the integrands must be equal in all coordinates r∈V:(7)$$\frac{\partial }{\partial t}C\left(r,t\right)=\nabla \left(D\left(r\right)\nabla C\left(r,t\right)-u\left(r\right)C\left(r,t\right)\right)$$

Distributed source. If, in addition, we have a distributed source or sink within the volume that is, respectively, generating or absorbing moles at a constant rate, an additional term can be added to the right-hand side of convection–diffusion differential Equation (6) as a distributed source/sink term R(r,t) (in mol⋅m^−3^⋅s^−1^). If the system interacts with the external world, modeled as exogeneous fluxes entering or leaving the system through the surface as a flux $j_{\mathrm{surf}}\left(r,t\right)$ (in mol⋅m^−2^⋅s^−1^), this can be added in the conservation equation as


(8)
$${\dot{N}}_{V}\left(t\right)=\frac{\partial }{\partial t}\int_{V}C\left(r,t\right)=\int_{\Omega }-j\left(r,t\right)\cdot \vec{n}\left(r\right)\;dr+\underset{=Q}{\underset{︸}{\int_{\Omega }-j_{\mathrm{surf}}\left(r,t\right)\cdot \vec{n}\left(r\right)\;dr}}$$


The concentration profile is not everywhere differentiable, this is especially so where the diffusion profile D(r) or the convection speed u(r) is discontinuous, which is the case at the interface between two different media. In this case, the continuity of the pressure and of the flux must be guaranteed at the interface, and the continuity equations yielding the appropriate interface boundary conditions are given as
(9)$$\{\begin{array}{c} \begin{array}{cc} \underset{\varepsilon \to 0^{+}}{\lim}j\left(r+\varepsilon ,t\right)-j\left(r-\varepsilon ,t\right)=0 & \mathrm{flux}\;\mathrm{continuity}\; \end{array}\\ \begin{array}{cc} \underset{\varepsilon \to 0^{+}}{\lim}\frac{C\left(r+\varepsilon \right)}{H_{+}}-\frac{C\left(r-\varepsilon \right)}{H_{-}}=0 & \mathrm{pressure}\;\mathrm{continuity} \end{array} \end{array}$$
where $H_{-}$ and $H_{+}$ are Henry’s constants of the medium before and after the interface, respectively. The flux continuity implies that there is no build-up of specifies on the interface, the pressure continuity transcribes the fact that the partial pressure of $CO_{2}$ depends on its species in the specific media, and the pressure must be in equilibrium across the interface.

In what follows, the concentration of the specific species will label with a superscript referring to the medium or the compartment in which it is dissolved, considering the following four compartments: blood (blood-irrigated tissues), skin (dermal tissue), meas (measurement cell), and coll (collection cell). In addition, we will solely consider the transport equations according to an axis orthogonal to the skin surface (which will be labeled as the z-axis) considering that all net fluxes in the (x,y)-plane at any given z-coordinate are null (constant concentration in the plane, at least in the proximity of the considered axis), unless explicitly introduced.

#### 2.1.1. Introducing the Four Compartments

When we refer to a compartment, we refer to a part of space in which the transporting medium is constant, which implies that the diffusion constant D is constant in that compartment. Four compartments will be distinguished: blood compartment, skin compartment, measurement cell and collection cell.

In the blood compartment, convection is determined by the net blood velocity, transporting the aqueous $CO_{2}$.

The skin compartment defines a more rigid structure that does not allow for convection [45,46,50]. This is a purely diffusive medium.

The measurement cell has no forced convective air flow (the transporting medium) but houses the measurement device, which is a dual-wavelength IR thermopile, the functioning of which is detailed below.

The collection cell has a convective air flow in order to avoid any building up of $CO_{2}$ concentration in the measurement cell. However, higher convection speeds, although allowing for faster dynamics, will result in a reduction in the gas’ concentration and hence a lower signal-to-noise ratio at the measurement device. In contrast, lower convection speeds will allow for build-up effects of the $CO_{2}$ concentration, resulting in an auto-regressive system favoring lower frequencies and, hence, slower dynamics.

#### 2.1.2. Measuring $CO_{2}$ Concentration with Dual-Wavelength IR Thermopiles

The detectors of the measurement cell consist of two thermopiles sensitive to incident photon energy. A measurement and reference thermopile are used with their optical filters centered around the nominal wavelengths $\lambda_{1}=4.26\;\mu m$ and $\lambda_{2}=3.91\;\mu m$, respectively. In order to reduce the contribution of other gases such as ambient air and water vapor, the NDIR differential measurement calculates the logarithm of the ratio of intensities at these two wavelengths.

$U_{\lambda_{1}}$ and $U_{\lambda_{2}}$ are the values read out from the thermopiles in the measurement cell in the presence of carbon dioxide, and $U_{0,\lambda_{1}}$ and $U_{0,\lambda_{2}}$ are the values read out from the thermopiles in the measurement cell in the absence of carbon dioxide. Considering the attenuation model of the Beer–Lambert’s law, this attenuation depends linearly on the molar concentrations of the constituents. With the chosen wavelengths $\lambda_{1}$ and $\lambda_{2}$, the attenuation measurement is a linear function of the molar concentration of carbon dioxide, independently of the concentration of water vapor. The attenuation profile $k_{{\mathrm{CO}}_{2}}$ of $CO_{2}$ at the two wavelengths is very discriminating, i.e., $k_{{\mathrm{CO}}_{2}}\left(\lambda_{1}\right)\gg k_{{\mathrm{CO}}_{2}}\left(\lambda_{2}\;\right)$, but not discriminative for water (H_2_O), i.e., $k_{H_{2}O}\left(\lambda_{1}\right)\approx k_{H_{2}O}\left(\lambda_{2}\right)$. We may thus write
(10)$$C_{CO_{2}}^{meas}\approx -\frac{1}{k_{CO_{2}}\left(\lambda_{1}\right)}\;\ln\left(\frac{U_{\lambda_{1}}/U_{\lambda_{2}}}{U_{0,\lambda_{1}}/U_{0,\lambda_{2}}}\right)$$

In order to account for the non-linear effects associated with the wavelength spectrum of IR light, the multiple lengths of light pathways due to light reflection and the non-linear response of the thermopile IR sensors, in Grangeat et al. [51,52], a linear-quadratic model using a non-integer power of gas concentration has been described in order to relate the thermopiles’ read-out voltage to the $CO_{2}$ concentration in the measurement cell. This model has been first proposed by Madrolle et al. for the quantification of a mixture of two diluted gases with a single metal oxide sensor (MOX) [53,54,55,56].

Knowing the logarithm of the voltage ratio, one can estimate the $CO_{2}$ molar concentration ($C_{CO_{2}}^{meas}{}_{{}_{}}$) in the measurement cell optical pathway using the following relationship (11) [53,54,55,56], where m, n, u are the (non-negative and real) quadratic model parameters.

If $\ell =ln\left[U_{0,\lambda_{2}}/U_{0,\lambda_{1}}\right]$ is not known, the intercept becomes another free parameter of the model:(11)$$-ln\left[U_{\lambda_{1}}/U_{\lambda_{2}}\right]\approx \ell +m\cdot {\left(C_{CO_{2}}^{meas}\right)}^{u}+n\cdot {\left(C_{CO_{2}}^{meas}\right)}^{2u}$$

We place ourselves within the framework of a supervised calibration with the existence of $N_{cal}$ samples of known composition $C_{CO_{2},i}^{meas}$. Using a Levenberg–Marquardt algorithm, the model parameters—m,n,u—are estimated by minimizing the following function Ψ:(12)$$\Psi =\overset{N_{cal}}{\underset{i=1}{\sum }}{\left[\ln\left[\frac{U_{\lambda_{1}}\left(i\right)}{U_{\lambda_{2}}\left(i\right)}\right]+\ell +m\cdot {\left(C_{CO_{2},i}^{meas}\right)}^{u}+n\cdot {\left(C_{CO_{2},i}^{meas}\right)}^{2u}\right]}^{2}$$

#### 2.1.3. Boundary Conditions for the System

The choice of boundary conditions is a prerequisite to solve the transport equations. In their article [57], Vaidya and Nitsche described the boundary conditions they have chosen for the simulation of the convection–diffusion equation of solutes in media with piecewise properties. In their article [36], Qazi et al. have described the boundary conditions they used to describe the kinetics of the carbon dioxide gas—liquid absorption on a membrane contactor.

At the interfaces, we will only conserve the diffusive flow, which means that the excess or default flow will leave or enter the system just before the interface.

For the direct model, we will consider the concentration at the extremal point of the known blood compartment (Dirichlet boundary condition). On the opposite side in the collection cell, we will suppose that there is no diffusive flow (zero derivative or von Neumann condition).

For the inverse model, we will consider the concentration obtained from the known thermopile voltage measurement (Dirichlet boundary condition) and suppose there is no diffusive flow at the extremal point of the blood compartment (von Neumann condition).

#### 2.1.4. Physical Parameters for the Transcutaneous $CO_{2}$ Transport

Figure 2 illustrates the characteristic parameters used in the 1D transport model. Note that in order to allow a better visualization of the figure, the relative scales along the z-axis do not correspond to the relative size of each compartments.

We have shown in this figure the inflow of blood. At the blood/skin interface, only the diffusive flux is conserved. We assume there is an outflow at the skin level within the blood to carry the convective flow, which does not cross the interface.

The air flow at the level of the collection cell allows mechanical convection to be established. We represent an incoming flow by the inflow arrows on both sides and an outgoing flow at the upper edge of the collection cell.

The variables and parameters of the transport model of carbon dioxide from the blood to the ambient air through the skin and the measuring device are defined in the following tables. The parameters are taken from the literature.

Length and width parameters of Table 1 and Table 2 are inherited from the device geometry. It defines the surface geometry, too.

Parameters defined in Table 1, Table 2 and Table 3 have been used to generate the data.

##### Physical Parameters for Blood Medium

The table’s symbols are outlined in the following equations:(13)$$u_{z}^{blood}=\frac{Q^{blood}}{A^{blood}}$$
(14)$$A^{blood}=\Delta x\cdot \Delta y$$
where:

$Q^{blood}$ is the blood flow (${\mathrm{cm}}^{3}\cdot s^{-1})$;$A^{blood}$ is the surface of the transverse section of the blood compartment $({\mathrm{cm}}^{2}$);$u_{z}^{blood}$ is the average blood velocity along the z axis ($\mathrm{cm}\cdot s^{-1})$;Δx is the length of the blood compartment (cm);Δy is the width of the blood compartment (cm).

**Table 1 sensors-23-06096-t001:** The characteristic physical parameters of the blood compartment at 42 °C.

Variable	Symbol	Measurement Unit	Value	Reference
Height	$\Delta z^{blood}$	cm	0.3	
Length	Δx	cm	5	
Width	Δy	cm	2	
Diffusion coefficient At 42 °C	$D^{blood}$	${\mathrm{cm}}^{2}\cdot s$ ^−1^	$2.2\cdot 10^{-5}$	Ghasem [58], Al Marzouqi [38], and Cao [59]
Average velocity	$u_{z}^{blood}$	cm·s ^−1^	$1.83\cdot 10^{-4}$	Formaggia [60], Vlachopoulos [61], and Guyton [62]
Henry coefficient	$H^{blood}$	adim		
Ostwald’s solubility coefficient of the carbon dioxide at 42 °C	$\beta^{blood}$	$\mathrm{mol}\cdot m^{-3}\cdot {\mathrm{mmHg}}^{-1}$	0.0275	
Temperature	$T^{blood}$	K	315.15	
Blood flow	$Q^{blood}$	${\mathrm{cm}}^{3}\cdot s$ ^−1^	$1.8\cdot 10^{-3}$	Chatterjee [16]
Blood surface	$A^{blood}$	${\mathrm{cm}}^{2}$	10	

Using the following formula, Formula (15), we compute the Henry constant for the blood based on the Ostwald’s solubility coefficient for the blood:
(15)$$H^{blood}=\beta^{blood}\;RT^{blood}$$
where:

R is the ideal gas constant—$R=0.0623637{\;m}^{3}\cdot \mathrm{mmHg}\cdot K^{-1}{\mathrm{mol}}^{-1}$;$\beta^{blood}$ is the Ostwald’s solubility coefficient of the carbon dioxide in the blood at 42 °C ($\mathrm{mol}\cdot m^{-3}\cdot {\mathrm{mmHg}}^{-1})$;$T^{blood}$ is the blood temperature (K).

We need to achieve an extrapolation to obtain the $\beta^{blood}$ Ostwald’s solubility coefficient of the carbon dioxide in the blood at 42 °C based on published values of the solubility coefficient given at 37 °C and 40 °C [3,33,63,64,65,66,67,68]:(16)$$\beta^{blood}=3.08\times 10^{-2}\;\mathrm{mol}\cdot m^{-3}\cdot {\mathrm{mmHg}}^{-1}\;\mathrm{at}\;37\;{}^{\circ}C$$
(17)$$\beta^{blood}=2.88\times 10^{-2}\;\mathrm{mol}\cdot m^{-3}\cdot {\mathrm{mmHg}}^{-1}\;\mathrm{at}\;40\;{}^{\circ}C$$
(18)$$\begin{array}{cc} \mathrm{Thus},\;\;\;\;\;\;\;\;\;\; & \\ & \beta^{blood}=2.75\times 10^{-2}\;\mathrm{mol}\cdot m^{-3}\cdot {\mathrm{mmHg}}^{-1}\;\mathrm{at}\;42\;{}^{\circ}C \end{array}$$

Finally, we obtain $H^{blood}=0.54$.

##### Physical Parameters for the Skin

We consider hereafter that the skin compartment corresponds to the stratum corneum, which has the main effect on the diffusion of the carbon dioxide from the blood compartment to the measurement cell through the skin.

Table 2 lists parameters used to characterize the skin. These are mean values, which are subject to intra- and inter-individual variations. For instance, skin elasticity might have an influence on skin compression induced by the wristband, which might have an influence on blood flow rate and average velocity.

**Table 2 sensors-23-06096-t002:** The characteristic physical parameters of the skin compartment at 42 °C.

Variable	Symbol	Measurement Unit	Value	Reference
Height	$\Delta z^{skin}$	cm	$1.6\cdot 10^{-3}$	
Length	Δx	cm	5	
Width	Δy	cm	2	
Diffusion coefficient	$D^{skin}$	${\mathrm{cm}}^{2}\cdot s$ ^−1^	$10^{-7}$	
Average velocity	$u_{z}^{skin}$	cm·s ^−1^	0	
Henry coefficient	$H^{skin}$	adim	1.6	Scheuplein [33]
Transcutaneous carbon dioxide flow rate per exchange surface area at 42 °C	$\frac{\Phi^{out\;skin}}{A}$	$\;\mathrm{nl}\cdot {\mathrm{cm}}^{-2}\cdot \min^{-1}$	290	
Bunsen carbon dioxide solubility at 42 °C	$\alpha_{CO_{2}}^{sc}$	$\frac{\mathrm{mL}\left(\mathrm{STPD}\right)/\mathrm{mLsolvent}}{\mathrm{mmHg}}$	$19.2\cdot 10^{-4}$	[33,64,65,66]
Skin conductance (volume flow rate) at 42 °C	$\frac{G_{diff\;pres}^{vol}}{A^{blood}}$	$\;\mathrm{mL}\cdot s^{-1}{\mathrm{cm}}^{-2}\cdot {\mathrm{mmHg}}^{-1}$	$121\cdot 10^{-9}$	Itoh [69]
Mass transfer coefficient	$k_{p}^{sc}$	$\;m\cdot s^{-1}$	$1.0\cdot 10^{-6}$	
Krogh’s diffusion constant	$K_{r}{}_{CO_{2}}^{sc}$	${\;\mathrm{cm}}^{2}\cdot s^{-1}\cdot {\mathrm{mmHg}}^{-1}$	$1.9\cdot 10^{-10}$	

The diffusion coefficient $D^{skin}$ is computed from Equation (19):(19)$$D^{skin}=\frac{K_{r}{}_{CO_{2}}^{sc}}{\alpha_{CO_{2}}^{sc}}$$
where:

$\alpha_{CO_{2}}^{sc}$ is the Bunsen carbon dioxide solubility in the stratum corneum (mL(STPD)/mL solvent/mmHg) (STPD stands for standard temperature, pressure, and dryness);$K_{r}{}_{CO_{2}}^{sc}$ is the Krogh’s diffusion constant (${\mathrm{cm}}^{2}\cdot s^{-1}\cdot {\mathrm{mmHg}}^{-1}$).

The Bunsen carbon dioxide solubility in the stratum corneum $\alpha_{CO_{2}}^{sc}$ is given by the following expression:(20)$$\alpha_{CO_{2}}^{sc}=\frac{H^{skin}}{P^{air}}$$
where:

$H^{skin}$ is the Henry coefficient of the skin;$P^{air}$ is the total air pressure at skin or membrane temperature.

If we suppose that $H^{skin}=1.6$ [33] and $P^{air}$ is 831.21 mmHg at 42 °C, we obtain:(21)$$\alpha_{CO_{2}}^{sc}=19.2\cdot 10^{-4}\frac{\mathrm{mL}\left(\mathrm{STPD}\right)/\mathrm{mL}\mathrm{solvent}}{\mathrm{mmHg}}\;\mathrm{at}\;42\;{}^{\circ}C$$

The Krogh’s diffusion constant is given by the following expression:(22)$$K_{r}{}_{CO_{2}}^{sc}=\frac{k_{p}^{sc}\cdot \Delta z^{skin}}{P^{air}}$$
where:

$k_{p}^{sc}$ is the mass transfer coefficient of the stratum corneum $\;(m\cdot s^{-1})$;$\Delta z^{skin}$ is the width of the stratum corneum (μm).

The mass transfer coefficient of the stratum corneum is computed from the conductance of the skin with respect to pressure difference per surface area using the following expression:(23)$$k_{p}^{sc}=\frac{G_{diff\;pres}^{vol}}{A^{blood}}P^{air}10^{-2}$$
where:

$G_{diff\;pres}^{vol}$ is the conductance of the skin with respect to pressure difference for the surface area of the device ($\mathrm{mL}\cdot s^{-1}\cdot {\mathrm{mmHg}}^{-1}$);$\frac{G_{diff\;pres}^{vol}}{A^{blood}}$ is the volumic conductance of the skin with respect to pressure difference per surface area ($\mathrm{mL}\cdot s^{-1}\cdot {\mathrm{cm}}^{-2}\cdot {\mathrm{mmHg}}^{-1})$;$A^{blood}$ is the exchange surface area (${\mathrm{cm}}^{2}$).

The conductance of the skin is computed from the flow rate per exchange surface area coming out of the skin in the open air:(24)$$\frac{G_{diff\;pres}^{vol}}{A^{blood}}=\frac{\Phi^{out\;skin}}{A^{blood}\left(P_{CO_{2}}^{blood}-P_{CO_{2}}^{ambient\;air}\right)\cdot 60\cdot 10^{6}}\approx \frac{\Phi^{out\;skin}}{A^{blood}P_{CO_{2}}^{blood}\cdot 60\cdot 10^{6}}$$
where:

$\Phi^{out\;skin\;}$ is the flow rate coming out of the skin ($\mathrm{nl}\cdot \min^{-1}$);$P_{CO_{2}}^{blood}$ is the carbon dioxide pressure in the blood (mmHg);$P_{CO_{2}}^{ambient\;air}$ is the carbon dioxide pressure in the ambient air (mmHg);$A^{blood}$ is the exchange surface area (${\mathrm{cm}}^{2}$).

Itoh et al. [69] gives the following values for the skin conductance (volume flow rate) per surface area $\frac{G_{diff\;pres}^{vol}}{A^{blood}}$ at the forearm level:

$77\cdot 10^{-9}\;\mathrm{mL}\cdot s^{-1}\cdot {\mathrm{cm}}^{-2}\cdot {\mathrm{mmHg}}^{-1}$ at 37 °C;

$130\cdot 10^{-9}\;\mathrm{mL}\cdot s^{-1}\cdot {\mathrm{cm}}^{-2}\cdot {\mathrm{mmHg}}^{-1}$ at 43 °C.

By linear interpolation, we compute the skin conductance (volume flow rate) per surface area $\frac{G_{diff\;pres}^{vol}}{A^{blood}}$ at 42 °C, which gives $121\cdot 10^{-9}\;\mathrm{mL}\cdot s^{-1}\cdot {\mathrm{cm}}^{-2}\cdot {\mathrm{mmHg}}^{-1}$.

If we assume the carbon dioxide pressure in the blood is 40 mmHg, and we neglect the carbon dioxide pressure in the ambient air, this corresponds to a flow rate per exchange surface area coming out of the skin in the open air $\frac{G_{diff\;pres}^{vol}}{A^{blood}}$ of 290 $\mathrm{nl}\cdot {\mathrm{cm}}^{-2}\min^{-1}$.

Below, we assume that the total air pressure $P^{air}$ at 42 °C is 831.21 mmHg, the Bunsen carbon dioxide solubility in the stratum corneum $\alpha_{CO_{2}}^{sc}$ at 42 °C is $19.2\;10^{-4}\frac{\mathrm{mL}\left(\mathrm{STPD}\right)/\mathrm{mL}\;\mathrm{solvent}}{\mathrm{mmHg}}$, and the height of the stratum corneum $\Delta z^{skin}$ is 16 μm.

Using the above expressions, we obtain the following values for the skin parameters:

$k_{p}^{sc}$ mass transfer coefficient of the stratum corneum: $1.0\cdot 10^{-6}m\cdot s^{-1}$;$K_{r}{}_{CO_{2}}^{sc}$ Krogh’s diffusion constant: $1.9\cdot 10^{-10}\;{\mathrm{cm}}^{2}s^{-1}{\mathrm{mmHg}}^{-1}$;$D^{skin}$ diffusion coefficient: $1.0\cdot 10^{-7}m^{2}\cdot s^{-1}$.

##### Physical Parameters for the Gaseous Media in the Column of the Capnometry Device

Typical value for the air flow is $1\;\mathrm{mL}/\min\;\left(i.e.,\;16.7\cdot 10^{-3}\;{\mathrm{cm}}^{3}\cdot s^{-1}\right)$. We varied this flow from 0.1 to 10 mL/min.

Typical value of $CO_{2}$ concentration in ambient air is 0.01613 µmol/cm^3^. We suppose here that a filter is placed in front of the air inlet of the collection cell, and this value is taken as null.

**Table 3 sensors-23-06096-t003:** The characteristic physical parameters of the gaseous medium in the measurement and the collection cell of the capnometry device at 42 °C.

Variable	Symbol	Measurement Unit	Value	Reference
Length	Δx	cm	5	
Width	Δy	cm	2	
Height	$\Delta z^{mes}$	cm	0.25	
	$\Delta z^{col}$	cm	0.25	
Diffusion coefficient at 42 °C	$D^{air}$	${\mathrm{cm}}^{2}\cdot s$ ^−1^	0.18	Qazi [36] and Eslami Faiz [39]
Average velocity	$u_{z}^{mes}$	cm·s ^−1^	0	
	$u_{z}^{col}$	cm·s ^−1^	1.67 · $10^{-3}$	
Henry coefficient	$H^{air}$	adim	1	
Air flow	$Q^{air}$	${\mathrm{cm}}^{3}\cdot s$ ^−1^	Typically, $16.7\cdot 10^{-3}$	
Carbon dioxide concentration in the ambient air	$C_{CO_{2}}^{in\;col\;air}$	$\mathrm{mol}/m^{3}$	0.01613	
Carbon dioxide pressure in the ambient air at 42 °C	$P_{CO_{2}}^{ambient\;air}$	mmHg	0.32	

### 2.2. A Discrete Space–Time CO_2_ Transportation Model

For an introduction to computer-aided modeling of material behavior, including physical and chemical parameters, and the mathematical tools for implementing those models in order to perform simulations, we refer to the referenced book of Jansens et al. [70] on computational materials engineering and, in particular, to chapter 5 by Kozeschnik on modeling solid-state diffusion [71]. The resolution of partial differential equations (PDE) using the finite difference method is described in the textbook by Langtangen and Ling [72]. Chapter 3 is dedicated to diffusion equations [73]. The textbook by Scherer on computational physics gives insight on numerical methods to compute physical models, in particular, for the discretization of differential equations [74].

We use a discrete state-space model in order to compute the time sequence of concentration profiles recursively along the transport path of the carbon dioxide, given a temporal sequence of carbon dioxide blood concentrations at the lower extremum of the blood compartment. This defines the recursive algorithm used for measurement data simulation; i.e., the previous time instant will provide the initial condition for the computation of the concentration profile along the transport path at the next time instant.

The following conventions for our notations are used: continuous scalar fields and constants will be denoted by light face characters. The space–time variables will appear as arguments between parentheses. Discrete representations will be given using arguments between square brackets; these quantities will appear in boldface lower-case characters when regrouped into a vector or boldface upper-case characters when regrouped into a matrix (a linear operator).

Table 4 lists the parameters and notations used.

#### 2.2.1. Spatial Discretization

In order to facilitate the discretization, we will use a finite difference scheme with regular, equidistant spatial sampling per compartment. At the boundaries we introduce a slack variable $p^{\mathrm{interface}}$ as
(25)$$p^{\mathrm{interface}}\overset{\mathrm{def}}{=}\underset{\varepsilon \to 0^{+}}{\lim}\frac{C_{-}\;\left(z-\varepsilon \right)}{H_{-}}=\underset{\varepsilon \to 0^{+}}{\lim}\frac{C_{+}\left(z+\varepsilon \right)}{H_{+}},$$
where the subscripts $\cdot_{-}$ and $\cdot_{+}$ refer, respectively, to the compartment below and above the interface (the z-axis pointing upwards). The slack variables can be eliminated from the equations using the interface conditions.

As per Figure 3, we have $N^{\mathrm{comp}}$ interior points for each compartment, together with two boundaries for which we have either a boundary or an interface condition, allowing to eliminate these from the set of unknowns. Given the thickness values of $\Delta z^{\mathrm{comp}}$ for the four compartments, we have the (relative) interface positions and may divide each compartment by placing $N^{\mathrm{comp}}$ equidistant points, which are separated from their neighbours by a distance of $\delta z^{comp}={\left(N^{\mathrm{comp}}+1\right)}^{-1}\Delta z$. This results in a position vector, $z={\left(z\left[1\right],\;z\left[2\right],\;\cdots ,\;z\left[n\right]\right)}^{T}$, where the superscript $\cdot^{T}$ is the transposition operator. If we note the concentration sampled in the ith point of the grid by $c\left[i\right]\left(t\right)=C\left(z_{i},t\right)$, we may form a concentration vector:
(26)$$c\left(t\right)={\left(\begin{array}{cccc} c\left[1\right]\left(t\right) & c\left[2\right]\left(t\right) & \cdots & c\left[n\right]\left(t\right) \end{array}\right)}^{T}$$

We define a stencil for the first-order and second-order spatial derivative operators within a given compartment determining the ith equation using the following expression:(27)$$\{\begin{array}{c} \frac{c\left[i+1\right]\left(t\right)-c\left[i-1\right]\left(t\right)}{z\left[i+1\right]-z\left[i-1\right]}=\frac{c\left[i+1\right]\left(t\right)-c\left[i-1\right]\left(t\right)}{2\;\delta z}\\ 4\frac{c\left[i+1\right]\left(t\right)-2\;c\left[i\right]\left(t\right)+c\left[i-1\right]\left(t\right)}{{\left(z\left[i+1\right]-z\left[i-1\right]\right)}^{2}}=\frac{c\left[i+1\right]\left(t\right)-2\;c\left[i\right]\left(t\right)+c\left[i-1\right]\left(t\right)}{{\left(\delta z\right)}^{2}} \end{array}$$

The above system of equations then gives the following expression:(28)$$\begin{array}{c} \frac{\partial }{\partial t}c\left[i\right]\left(t\right)=\left(\frac{D}{{\left(\delta z\right)}^{2}}+\frac{u}{2\delta z}\right)\;c\left[i-1\right]\left(t\right)-\frac{2D}{{\left(\delta z\right)}^{2}}\;c\left[i\right]\left(t\right)+\left(\frac{D}{{\left(\delta z\right)}^{2}}-\frac{u}{2\delta z}\right)\;c\left[i+1\right]\left(t\right) \end{array}$$

We may thus express the spatial discretization of the convection–diffusion equation under matrix form:(29)$$\frac{\partial }{\partial t}c\left(t\right)=F\;c\left(t\right)$$
where F is a linear finite-difference operator that encodes the spatial derivatives as well as the boundary and interface conditions defined below.

Firstly, Dirichlet boundary condition is applied on the lower boundary. Then, the only equation that depends on the exogenous (boundary value) will be the one at index i=1. We will have a new term $\left(\frac{D}{{\left(\delta z\right)}^{2}}+\frac{u}{2\delta z}\right)c^{\mathrm{boundary}}\left(t\right)$ replacing the term that depends on c[0](t).

Secondly, Neumann boundary condition is imposed on the upper boundary (equation of index n). The condition states that there exists a point beyond the boundary such that
$$\frac{c^{\mathrm{exo}}\left(t\right)-c\left[n-1\right]\left(t\right)}{2\;\delta z}=0\;\overset{}{\Leftrightarrow }\;c\left[n-1\right]\left(t\right)=c^{\mathrm{exo}}\left(t\right)$$ Hence, the nth equation reads
(30)$$\begin{array}{c} \frac{\partial }{\partial t}c\left[n\right]\left(t\right)=\frac{2D}{{\left(\delta z\right)}^{2}}\;c\left[n-1\right]\left(t\right)-\frac{2D}{{\left(\delta z\right)}^{2}}\;c\left[n\right]\left(t\right) \end{array}$$

And finally, Robin boundary condition is applied at each interface between model compartments. This boundary condition is imposed on a virtual interface point situated in between the ith and i+1th sample points. We have the interface conditions that are built on one-sided differences in order to consider a single medium. At the interface, we have the limit concentrations in both media that are, respectively, given by $c_{-}^{\mathrm{interface}}$ and $c_{+}^{\mathrm{interface}}$ before and after the interface:(31)$$\begin{array}{c} D_{-}\frac{c_{-}^{\mathrm{interface}}-c\left[i\right]}{\delta z_{-}}-u_{-}\;c_{-}^{\mathrm{interface}}=D_{+}\frac{c\left[i+1\right]-c_{+}^{\mathrm{interface}}}{\delta z_{+}}-u_{+}\;c_{+}^{\mathrm{interface}}\\ \frac{c_{-}^{\mathrm{interface}}}{H_{-}}=\frac{c_{+}^{\mathrm{interface}}}{H_{+}}=p^{\mathrm{interface}} \end{array}$$ From this, we achieve:(32)$$c_{-}^{\mathrm{interface}}=\frac{\frac{D_{+}}{\delta z_{+}}c\left[i+1\right]+\frac{D_{-}}{\delta z_{-}}c\left[i\right]}{\frac{D_{-}}{\delta z_{-}}-u_{-}+\frac{H_{+}}{H_{-}}\left(\frac{D_{+}}{\delta z_{+}}+u_{+}\right)}$$

This can be re-injected into the convection–diffusion equation, analogously:(33)$$\begin{array}{c} c_{+}^{\mathrm{interface}}=\frac{H_{+}}{H_{-}}c_{-}^{\mathrm{interface}} \end{array}$$

If the flow is purely diffusive through the interface, we have
(34)$$\begin{array}{c} c_{-}^{\mathrm{interface}}=\frac{\frac{D_{+}}{\delta z_{+}}c\left[i+1\right]+\frac{D_{-}}{\delta z_{-}}c\left[i\right]}{\frac{D_{-}}{\delta z_{-}}-\frac{H_{+}}{H_{-}}\;\frac{D_{+}}{\delta z_{+}}} \end{array}$$

#### 2.2.2. Temporal Discretization

In order to integrate the above equation, we use an implicit Euler scheme [73] (chapter 3.2), [75,76] in which temporal sampling at regular intervals of δt leads to
(35)c(t+δt)−c(t)=δt F c(t+δt) ⇔ (Idn−δt F)c(t+δt)=c(t)
where Idn is the identity operator on ℝn.

The discretized scheme is thus given by
(36)$$A\;c_{k+1}=c_{k}$$
where the index k is the time sample index linked to the wall clock time, $t_{0}+\left(k-1\right)\delta t$ and $A\overset{\mathrm{def}}{=}\left({\mathrm{Id}}_{n}-\delta t\;F\right)$ is the implicit linear operator. Since all eigenvalues of F are strictly negative, the matrix A has eigenvalues that are strictly bigger than one; hence, its inverse is a contractive operator, resulting in the stability of the algorithm. Note that to differentiate *t* (time) and *k* (time sample index), we use subscript for *k* and square brackets for [*t*].

#### 2.2.3. Exogenous Inputs

To cope with exogeneous inputs in the system, such as the blood transport flow or the forced air flow through the collection cell, the equation is augmented with a steering matrix G and the exogenous inputs q that encode the boundary conditions as well as the flows:(37)$$\frac{\partial }{\partial t}c=Fc+Gq$$

In discrete form, this gives
(38)$$c_{k+1}-c_{k}=\delta t\;\left(F\;c_{k+1}+G\;q_{k+1}\right)\;\;\;\Leftrightarrow \;\;\;\;A\;c_{k+1}=c_{k}+\delta t\;G\;q_{k+1}$$

G and q carry the information about the Dirichlet boundary condition at the lower bound, which gives
(39)$$G\left[1,\;1\right]=\frac{D}{{\left(\delta z\right)}^{2}}+\frac{u}{2\delta z}$$
and
(40)$$q\left[1\right]=c_{k}^{\mathrm{boundary}}$$

#### 2.2.4. State-Space Model and Noise

We define $y_{k}$ as the observation vector describing the measurement in the measurement cell at the time sample index *k*. As there is only one measurement, the vector $y_{k}$ is a scalar $y_{k}$. The observation equation is
(41)$$y_{k}=h^{T}c_{k}+v_{k}\;$$
where:

h is a canonical vector for which sole non-zero entry is associated with the position of the NDIR optical measurement (the last grid point in the measurement cell before the interface with the collection cell);$v_{k}\sim N\left(0,\sigma_{v}^{2}\right)$ is the observation noise with a variance–covariance matrix R that describes the inaccuracies of sensor outputs as measurements are taken.

The difficulty in the inversion of the model is that no boundary condition can replace the knowledge of the concentration at the blood level. To overcome this difficulty, we suppose that the variations in concentration are slow and correspond to a first-order auto-regressive process, i.e.,
(42)$$\frac{\partial }{\partial t}C\left(z_{\mathrm{blood},\mathrm{in}},t\right)=\varphi \;C\left(z_{\mathrm{blood},\mathrm{in}},t\right)+w^{in\;blood}\left(t\right)$$
where:

$w^{in\;blood}\left[t\right]$ is the input noise model used to generate the input signal $C\left(z_{\mathrm{blood},\mathrm{in}},t\right)$; it is a white Gaussian noise process of variance $\sigma_{w}^{2}$.φ is a signal regularity parameter of a first-order autoregressive signal model.

To derive the state-space equation, we combine this input signal model with the physical model describing the transport of the carbon dioxide. The associated state-space equations in discrete form read
(43)$$A\;c_{k+1}=c_{k}+\delta t\cdot G\;q_{k+1}+\delta t\cdot w_{k+1}$$
where $\delta t\cdot w_{k+1}\sim N\left(0,Q\right)$ is the noise process that integrates the modeling errors. N(0,Q) is a normal vector distribution, with each component of zero average, and with a variance–covariance matrix Q.

We assume that the modeling error noise w[t] and the observation noise v[t] are independent.

### 2.3. Estimation of $CO_{2}$ Blood Concentration Using Kalman Filter

#### 2.3.1. Kalman Filter Algorithm

The resolution of mathematical equations associated with diffusion phenomena is described in the reference book by Crank [77]. But in this book, the numerical methods do not consider noisy observations or state evolutions. Our problem is to estimate the state vector $c_{k,k}$ based on the previous state vector $c_{k-1,k-1}$ and the (noisy) measurement $y_{k}$. We propose to use a Kalman filter, which is an optimal filter that recursively computes a linear, least-mean square estimator [24,25,26]. The Kalman filter is adaptative in the sense that it tracks the noise level of both the measurements and the state evolution equations by updating the noise covariance matrices at each temporal step.

The Kalman algorithm works in a “prediction-correction” loop as described below and in [51,78]. For this description, we use the notations defined in Table 5.

Algorithm initialization:

 During the initialization phase, at the time step k=0, $c_{0,0}$ is initialized to null vector, and $P_{0,0}$ is initialized to identity matrix.

2.Prediction step: Extrapolation (prediction) of the state and uncertainty covariance matrix at time step k+1 from the current state (at time step k). We present below the equations for the implicit method used for time integration scheme. Equation (44) is for state extrapolation, and Equation (45) is for covariance extrapolation:


(44)
$$\underset{A}{\underset{︸}{\left(I-F\cdot \delta t\right)}}c_{k+1,k}=c_{k,k}+Gq_{k+1}\cdot \delta t$$



(45)
$$P_{k+1,k}=A^{-1}P_{k,k}A^{-1}{}^{T}+Q$$


3.Correction step: State vector and uncertainty covariance matrix update using the estimates computed at time step k−1, $c_{k,k-1}$ and $P_{k,k-1}$, and the current measurement $y_{k}$.

 The Kalman gain given in Equation (46) expresses the weights given to the new noisy measurement $y_{k}$ or the estimated $c_{k,k-1}$ state, which is also subject to different external disturbances:


(46)
$$K_{k}=P_{k,k-1}h^{T}{\left(hP_{k,k-1}h^{T}+R\right)}^{-1}$$


 The estimation of the current state (a posteriori state estimator) of the system is performed by taking a linear combination between the estimate made at the previous moment k−1 (a priori state estimator) and the new recorded data, as shown in Equation (47) below. Knowing the Kalman $K_{k}$ gain expression, we can give the expression to estimate the concentration $c_{k,k}$ at time k based on the difference between the current measurement $y_{k}$ and the estimated measurement $hc_{k,k-1}$ from the previous state estimate. This difference defines the prediction error ($y_{k}-hc_{k,k-1}$), which evaluates the amount of new information brought by the current measurement:


(47)
$$c_{k,k}=c_{k,k-1}+K_{k}\left(y_{k}-hc_{k,k-1}\right)$$


 The estimation of the covariance matrix of the estimate error at time k from the estimated covariance matrix $P_{k,k-1}$ at time k−1 is based on Equation (48):


(48)
$$P_{k,k}=\left(I-K_{k}h\right)P_{k,k-1}{\left(I-K_{k}h\right)}^{T}+K_{k}RK_{k}^{T}$$


4.End of iteration:

 The steps are iterated until either the end is decided by the user or the measurements are stopped. The output of the algorithm is the estimated $CO_{2}$ blood concentration corresponding to the first element of vector $c_{k,k}$, at each time step.

#### 2.3.2. Estimation of $CO_{2}$ Blood Pressure

Physicians are using $CO_{2}$ blood pressure $P_{CO_{2}}^{in\;blood}$ as a variable to characterize $CO_{2}$ blood concentration $C_{CO_{2}}^{in\;blood}$. This pressure is defined by the Henry law as the pressure the carbon dioxide would have in an air gas phase in equilibrium with the blood liquid phase:(49)$$P_{CO_{2}}^{in\;blood}=\frac{C_{CO_{2}}^{in\;blood}}{\beta^{blood}}$$
where $\beta^{blood}$ is the Ostwald solubility coefficient of the carbon dioxide in the blood.

In this article, we are considering the carbon dioxide blood concentration variable.

### 2.4. Methods of Evaluation

The evaluation is conducted in three steps.

Firstly, we evaluate the simulation of the direct transport problem. To model the $CO_{2}$ desorption for the four modelized compartments (blood, skin, measurement and collection), we used a minimum number of discretization points, 3 per compartment, plus the outer interface: N=13. The following results are simulated using the parameters presented in Table 1, Table 2 and Table 3. The $C_{CO_{2}}^{in\;blood}$ input signal is simulated as a step transition between a normocapnia level and a hypercapnia level. The normocapnia level is defined as an average concentration (μ = 1.099 mol/m^3^) of $CO_{2}$_,_ which corresponds to a pressure of 40 mmHg.

Secondly, the results obtained for the transport inversion problem are illustrated on the previous simulations with different levels of noise (Gaussian zero mean white noise) added to the simulated observation vector to test the ability of the Kalman filter to adapt to the noise level.

Finally, we simulate a more realistic clinical test case simulating a sequence of stages with hypocapnia, normocapnia and hypercapnia levels. The Kalman inverse problem results are simulated.

The optimization of the numerical processing for the inversion problem is related to the quantification of the ${\tilde{C}}_{CO_{2}}^{in\;blood}$ $CO_{2}$ concentration level estimated in the blood medium. In these simulations, we assume that the $CO_{2}$ concentration in the incoming air is zero: $C_{CO_{2}}^{in\;col\;air}=0$. This is equivalent to assuming that a carbon dioxide filter has been placed over the air flow at the inlet of the collection cell.

The results are evaluated according to several performance factors with respect to the quantification of the $CO_{2}$ blood concentration estimated and to the time necessary for this quantification.

First performance parameter is calculated as the relative difference between the mean on the hypercapnia level (direct problem $\mu_{hypercapnia}$) for input concentration $C_{CO_{2}}^{in\;blood}$ and the mean of the hypercapnia level (inverse problem ${\hat{\mu }}_{hypercapnia}$) of the estimated input concentration ${\tilde{C}}_{CO_{2}}^{in\;blood}$. The hypercapnia means are calculated beyond the time when the level difference with the initial level (normocapnia) has reached 90% of the difference between the level at equilibrium after the transition (hypercapnia) and the initial level of normocapnia. In case of the “clinical” test, the evaluation is performed in the same way on the different successive capnia levels simulated (hypo, hyper1, hyper2 and normo capnia).

The second parameter is the root of the mean square error (RMSE) between the $C_{CO_{2}}^{in\;blood}$ concentration in the blood as input signal and the ${\tilde{C}}_{CO_{2}}^{in\;blood}$ estimated concentration in the blood as output signal. Computation of the RMSE is illustrated in Figure 4a. First signals are aligned, and then the root of the mean square error is calculated on the aligned signals.

The rise time ($t_{r}$) corresponds to the difference between the time necessary for the initial level (normocapnia) to reach, respectively, 90% and 10% of the difference between the level at equilibrium after the transition (hypercapnia) and the initial level of normocapnia, and it is of interest for the temporal performance of the sensor. It is computed on the estimated concentration ${\tilde{C}}_{CO_{2}}^{in\;blood}$. The method of rise time calculation is illustrated in Figure 4b. In the case of a “clinical” test, the rise time calculation is performed between the successive different capnia levels; indeed, some of them are fall time (normo to hypo or hyper to normo).

Other temporal performance parameters are the delays corresponding to direct, inverse and global problems. They are computed by analyzing the global temporal profile of the concentration along the transport of the carbon dioxide after placing a concentration step variation as input, as illustrated in Figure 4a.

The time difference corresponding to the maximum of the cross-correlation between the pulse at the input and the pulse at the level of the measurement cell defines the direct delay time ($t_{d-dir}$). This delay time is comparable to a propagation time of the carbon dioxide in the blood, the skin and the device up to the measurement cell. The time difference corresponding to the maximum of the cross-correlation between the pulse at the level of the measurement cell and the pulse at the level of the blood cell after the inverse problem defines the inverse delay time ($t_{d-inv}$). The time difference corresponding to the maximum of the cross-correlation between the pulse at the input and the pulse at the level of the blood cell after the inverse problem defines the global delay time ($t_{d-global}$).

These previous performance factors allow us to estimate two global performance factors. The relative performance ($perf_{rel})$ is defined as the ratio of the difference between the mean of the estimated and the input signal concentrations at the hypercapnia level. The global signal-to-noise ratio $\left(rsb_{global}\right)$ is defined as the root mean square of the input signal and the RMSE:(50)$$perf_{rel}=100\times \left(\mu_{hypercapnia}-{\hat{\mu }}_{hypercapnia}\right)/\mu_{hypercapnia}\;\;\;\;\;\;\;\;\;\;\left(\%\right)$$
(51)$$rsb_{global}=20\times \log10\left(rms\left(C_{CO_{2}}^{in\;blood}\right)\right)/rmse\;\;\;\;\;\;\;\;\;\;\;\;\;\left(\mathrm{dB}\right)$$
where $rms\left(C_{CO_{2}}^{in\;blood}\right)$ is the quadratic norm of the input signal.

#### 2.4.1. Evaluation of the Direct Approach

For all simulations, the input signal for our direct approach is illustrated in Figure 5. The input $CO_{2}$ concentration value $C_{CO_{2}}^{in\;blood}=1.099\;\mathrm{mol}/m^{3}$ is the average value (μ) of the concentration in the blood, which corresponds to a pressure of $P_{CO_{2}}^{in\;blood}=40\;\mathrm{mmHg}$. Hypercapnia level is defined as an increase of 10 mmHg, $P_{CO_{2}}^{hypercapnia}=50\;\mathrm{mmHg}$, which corresponds to a concentration of $C_{CO_{2}}^{hypercapnia}=1.3738{\;\mathrm{mol}/m}^{3}$.

#### 2.4.2. Evaluation of the Complete Model for $CO_{2}$ Quantification

We consider three different noise variances added to the direct simulation result in order to see how they impact the time and to illustrate the adaptive property of the Kalman filter with respect to the noise level. We will also quantify the mean of the estimated hypercapnia blood level. We will examine the root mean square error between the real value of $CO_{2}$ blood concentration and the one estimated by the algorithm. The input signal of the Kalman filter $C_{CO_{2}}^{meas}$ is the result of the direct problem applied to a $C_{CO_{2}}^{in\;blood}$ step function with no noise, low noise and high noise added, presenting, respectively, a variance of $0\;{\left(\mathrm{mol}/m^{3}\right)}^{2},\;10^{-8}\;{\left(\mathrm{mol}/m^{3}\right)}^{2}\;\mathrm{and}\;10^{-6}\;{\left(\mathrm{mol}/m^{3}\right)}^{2}$), as illustrated in Figure 6.

#### 2.4.3. Evaluation on a Realistic Simulated Clinical Case

In [41], Pierre Grangeat et al. described results of a clinical test conducted with a previous version of the CAPNO capnometry wristband in which the convection in the device channel was only induced by a difference in temperature between the ambient air and the heated skin, and the collection cell was under the measurement cell. We simulate such a test by considering a hypocapnia (decrease of 10 mmHg) phase, followed by a normocapnia phase and two different levels of hypercapnia (two successive increases of 5 mmHg), and then returning to the normocapnia condition. The objective of this test is to simulate a realistic experiment as it was conducted during a previous clinical evaluation. The simulation model is slightly different from the model described above. It is written using a compartmental model as described in [79].

The capnia levels and the duration of each phase are given in Table 6 and illustrated in Figure 7. We add two phases to the clinical chronogram:
An initialization phase to simulate the time that was used in the clinical test after installation of the device on the patient and before the test recording. During this initialization phase, we simulate a normocapnia level.An extension phase in order to observe on simulations the return to normocapnia equilibrium. During this extension phase, we simulate a normocapnia level.

**Table 6 sensors-23-06096-t006:** Chronogram of capnia phases of a realistic clinical simulation.

Phase Name	Phase Start Time (s)	Duration (s)	$\mathrm{Capnia}\;\mathrm{Level}\;(mol/m^{3})\;/\;mmHg$
Initialization	1	1188	1.099/40
Normocapnia 1	1189	374	1.099/40
Hypocapnia	1563	240	0.824/30
Normocapnia 2	1803	531	1.099/40
Hypercapnia 1	2334	375	1.236/45
Hypercapnia 2	2709	368	1.374/50
Normocapnia 3	3077	587	1.099/40
Extension (end of measurement)	3664	990	1.099/40
End of simulation	4654		

**Figure 7 sensors-23-06096-f007:**
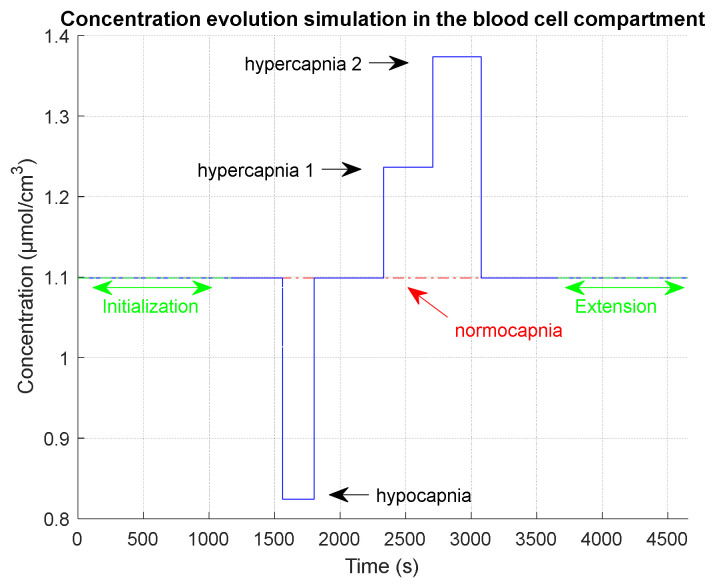
Simulation scheme for a realistic clinical test with different phase durations and levels, as given in Table 6.

Figure 7 shows the capnia phase sequencing in the input of the model after an initialization phase in which equilibrium is reached and with an extension phase to observe the complete return to equilibrium. The hypocapnia phase lasts about 4 min, and the hypercapnia phases last about 6 min each.

We simulated realistic clinical data and examined the performance parameters.

## 3. Results

### 3.1. Performance Parameters for the Direct Model

Figure 8 illustrates in the four compartments the propagation of $CO_{2}$ through the different media according to the convection–diffusion model described above.

The time necessary for the system initialization is visible in Figure 8a. For all other figures in the Results section, the initialization phase is omitted, as in Figure 8b. The figures are presented after the time necessary for the system initialization (i.e., after the time necessary to reach the normocapnia level as the system is initialized to 0).

The optimization of the device time response involves the adjustment of certain parameters such as the ambient air flow entering the collection cell, which is responsible for the mechanical convection. This is necessary to extract the transcutaneous $CO_{2}$ flow, preventing the accumulation of $CO_{2}$ inside the device. An analysis related to the air flow traversing the collection cell is necessary. In a first intention, we will vary the ambient air flow between $10^{-1}\;\mathrm{mL}/\min$ and 10 mL/min in order to obtain a partial pressure variation equal to the partial pressure associated with carbon dioxide in the ambient air.

The ambient air flow entering the collection cell contributes to the decrease in the response time of the system as well as a reduction in propagation time through the transport column but, in return, dilutes the concentration in the measurement and in the collection cell and, therefore, the level of the signal. There is a compromise to be made between lower time constants and the signal level at the output of the collection cell, as shown on Figure 9.

### 3.2. Performance Parameters for the Inverse Model

We tested the Kalman inversion algorithm on the result of the direct problem. We present the results successively for the three noise levels. In these results, the Kalman filter has been initialized to zero. The model noise variance is set to $10^{-8}\;{\left(\mathrm{mol}/m^{3}\right)}^{2}$, while the observation noise variance is set to $10^{-6}\;{\left(\mathrm{mol}/m^{3}\right)}^{2}$. The signal regularity parameter is φ = 0.

#### 3.2.1. Noiseless Observation of a Step Function as Input

Firstly, we add no noise variance to the measured simulated output signal of the measurement cell. In Figure 10a, the Kalman filter estimation of the propagation of $CO_{2}$ through the different media in the four compartments is illustrated.

The comparison of the $C_{CO_{2}}^{in\;blood}$ signal at the simulation input (curve in black) versus the estimated ${\tilde{C}}_{CO_{2}}^{in\;blood}$ signal (curve in red) is illustrated on Figure 10b.

#### 3.2.2. Low Noise Observation of a Step Function as Input

In the second test, we add a low noise variance to the measured simulated output signal of the measurement cell (cyan curve). Similarly, Figure 11a illustrates the propagation of $CO_{2}$ through the different media in the four compartments.

The comparison of the $C_{CO_{2}}^{in\;blood}$ signal at the simulation input (curve in black) versus the estimated ${\tilde{C}}_{CO_{2}}^{in\;blood}$ signal (curve in red) is shown in Figure 11b.

#### 3.2.3. High Observation Noise of a Step Function as Input

Finally, in the third test, we add a high noise variance to the measured simulated output signal of the measurement cell (cyan curve). Similarly, Figure 12a illustrates the propagation of $CO_{2}$ through the different media in the four compartments.

The comparison of the $C_{CO_{2}}^{in\;blood}$ signal at the simulation input (curve in black) versus the estimated ${\tilde{C}}_{CO_{2}}^{in\;blood}$ signal (curve in red) is shown in Figure 12b. The estimated ${\tilde{C}}_{CO_{2}}^{in\;blood}$ curve is slightly noisy, but the mean hypercapnia level is well estimated.

#### 3.2.4. Performance Parameters

The results obtained analyzing the direct and inverse approach are presented in Table 7.

The comparison of the $C_{CO_{2}}^{in\;blood}$ signal at the simulation input (curve in black) versus the estimated ${\tilde{C}}_{CO_{2}}^{in\;blood}$ signal (curve in red) is illustrated in Figure 10b for the noiseless observation vector, in Figure 11b for low noise and in Figure 12b for high noise. We observe the delay to reach the hypercapnia measurement on each of these curves. For all noise levels tested, the hypercapnia level is reached.

These curves show the adaptability of the Kalman filter to react to the noise added to the signal. In all cases, the hypercapnia level is recovered. The Kalman filter has the advantage of acting as a real filter for the noise-to-signal ratio.

Delays and rise times are of the same order; they cannot be shorter than the physiology. Furthermore, the rise time of the global model is of the same order as the direct one, which makes our choice of a recursive Kalman filter as an inversion algorithm a realistic choice for a real-time device.

### 3.3. Performance Parameters for the Complete Model on a Realistic Clinical Test

Previously, we showed the adaptability of the Kalman filter for observation noise. Our purpose here is to show its adaptability to the model noise.

#### 3.3.1. Compartment Model Simulation

The result of the simulation with the compartment model is illustrated in Figure 13. The durations of the different capnia levels are short; nevertheless, we observe the different levels on the observation curve (measurement cell).

The Kalman inversion of the realistic clinical data simulation produces the result presented in Figure 14. This calculation is performed with (a) regularity parameter φ=0 and (b) regularity parameter φ=−0.0036.

#### 3.3.2. Compartment Model Simulation

Figure 15 presents detailed comparisons, firstly, in the blood cell, between blood $CO_{2}$ concentration in the input of the simulation model and the estimated $CO_{2}$ concentration using the Kalman filter (right scale), and, secondly (left scale), of the simulated measured $CO_{2}$ concentration in output of the simulation model with added noise (variance of $10^{-6}\;{\left(\mathrm{mol}/m^{3}\right)}^{2})$ and the estimated $CO_{2}$ concentration in the measurement cell after Kalman inversion.

As we can see, the φ parameter allows us to compensate for concentration discrepancy in the blood cell. Magenta and red curves are on the same level in Figure 15b. However, we observe that the hypocapnia and hypercapnia levels are underestimated. The underestimation of the hypocapnia level is mainly due to the fact that the duration of the hypocapnia phase is too short.

#### 3.3.3. Performance Parameters

The results are analyzed considering the six performance parameters defined above. They are presented in Table 8 depending on the parameters and presented in Table 9 for the global parameters, specifically.

The main result of comparing the two columns of these tables according to the regularity parameter φ is that we see how the regularity parameter can act on the recovery of capnia levels. The relative performance parameter decreases from around 250% to less than 5%.

It shows also that, with this open chamber configuration (with forced convection), we can observe capnia phases of approximately 6 min time length, which is less than 10 min.

In [10], Dervieux et al. used a closed chamber model. They stated that if the sensor has a height of 1 mm, a 95% response will be achieved after 1 h 35 min and that, therefore, for a sensor to have a reasonable response time—e.g., below 10 min—it must be relatively thin, specifically in the range of 100 µm. We show in this work that, using an open chamber principle, this conclusion on the thickness of the device can still be released. Our simulated device has a thickness of 0.5 cm (measurement and collection cell thickness). This makes an open chamber device an alternative to a thin-film patch proposition.

## 4. Discussion

In this work, we propose an open chamber principle with a continuous circulation of air flow. But this requires the development of a dynamic model of carbon dioxide transport through the skin to recover the carbon dioxide blood content from the measurement sequence in the measurement cell. This allows the design of a model-based recursive signal processing approach based on a Kalman filter for a real-time estimation of the carbon dioxide blood pressure. A Kalman filter is relevant to estimate hidden variables in a noisy environment. The processing can be implemented with limited computational and memory resources.

This dynamic model described the transport of carbon dioxide from the blood to the collection cell through the skin and the measurement cell. It is based on convection–diffusion equations in those compartments.

This model states all the parameters that have an influence on the measurement. These include parameters linked to the device architecture, to the IR source and sensors, to the operating mode, or to the patients. In Section 3, we illustrated the influence of the ambient air flow rate on the device time response and on the measurement level. Temperature also has an influence on both the physiological parameters of the model, such as blood velocity and blood flow, and on physical parameters such as the Henry coefficients, the solubility coefficients, the skin conductance and the associated mass transfer coefficient and Krogh’s diffusion constant, the diffusion coefficients, and the carbon dioxide concentration and pressure in the ambient air. In this article, we have worked with a constant temperature of 42 °C. We have used mean values for skin parameters.

The variability of these parameters might have an influence on the measurement variability, including the intra-measurement variability linked to patient non-stationarity or measurement noise, inter-measurement variability for the same patient but for different operating modes or different devices, and inter-measurement variability among different patients.

The operating mode should also be well established. For instance, before starting the measurement, the device should be set on the patient, and some time should pass before starting the measurement in order to reach an initial equilibrium among the carbon dioxide contents in the four compartments.

The study of the measurement sensitivity to all these variabilities needs specific trials on experimental benches or on clinical trials. This should be performed to control the reliability of the measurement. For the parameters that are the more sensitive, complementary measurements should be achieved. This might be relevant, for instance, for patient-specific parameters including blood flow; skin thickness; skin vascularization; skin gas emission, such as water vapor induced by sweat or other volatile organic compounds (VOC); and contact between the device and the patient to control the skin heating or the gas loss according to the sealing of the carbon dioxide flow between the skin and the device. It might also be relevant for operating modes such as the variability of the ambient air parameters, including the carbon dioxide concentration, temperature pressure, humidity level, and flow speed. The measurement robustness is of primary interest for the development of a wearable device to be used in a homecare environment on moving people.

The Kalman filter framework associated with the state-space model allows us to take into account the variability. In the simplest version, the variability is described by the two noise variables included in the model: the model noise, which might describe the errors linked by the use of an approximate state-space model, and the observation noise of the measurement error, including the inaccuracy error. The Kalman filter allows a recursive estimation of the variance of the noises and, thus, an improvement of the robustness with respect to measurement variability.

Parameters linked to the device architecture might be controlled by relevant calibration. The superiority of the IR measurement with respect to electro-chemical or electro-optical measurements is that frequent sensor calibration should not be required.

We have described here a simplified 1D version with a transport of the carbon dioxide gas along the axial direction perpendicular to the skin surface. The next step will be a two-dimensional (2D) version to take into account, using the same model, the transverse direction parallel to the skin surface and the axial direction. This 2D model will also allow us to consider a device architecture based on a transverse air convection flow parallel to the skin surface.

## 5. Conclusions

The development of a wearable device for the continuous monitoring of the blood carbon dioxide content is of primary interest for a homecare environment. It belongs to the large category of wearable health devices for vital sign monitoring described by Dias and Silva Cunha [80]. It belongs also to the general trend towards developing skin-based wearable devices as described by Jin et al. [81], wearable light sensors based on graphene material as described by Akinwande and Kireev [82,83], and wearable and miniaturized sensors for personalized and preventive medicine as described by Tricoli et al. [84]. The development of new room-temperature gas sensors based on a metal–organic framework nanocomposite as proposed by Zhang et al. [85] also gives a new perspective for using a gas sensor on a wristband. Tomasic et al. [86] have explained that the development of a wearable capnometry device is a requirement for the continuous monitoring of a COPD patient.

However, the development of such a capnometry device is a challenging topic since the transcutaneous carbon dioxide flow is very low: in the range of 0 to 800 $\mathrm{nL}/{\mathrm{cm}}^{2}/\min$.

Therefore, we have proposed in this paper a dynamic model of carbon dioxide transport through the skin using a capnometry wristband. Such a dynamic model is a requirement not only for designing accurate instrumentation but also for computing simulated data and developing model-based signal processing.

Thanks to this model, we have studied a new architecture in which the measurement cell is in contact with the skin and a continuous air flow is collecting the carbon dioxide gas. In this paper, we have proved with simulated data the feasibility of such a technological concept.

Such a model is a key contribution for the development of an autonomous wearable device as discussed in Section 4. The signal processing should be embedded to lower the data exchange rate between the wearable device and the smart phone or the computer that should store the data. It should also be resilient to measurement variability factors.

To improve the autonomy of the device, we have worked, in this paper, on the model to embed the data processing. But further studies will be required to lower the electrical power consumption. The main electrical power requirements are linked to data communication, skin heating and IR light emission.

We have described, in this paper, the transport of carbon dioxide from the blood to the collection cell. But such a model can be extended to other volatile molecular species, such as volatile organic compounds (VOC), including ethanol, acetone, isoprene or methane, and other diluted blood gases such as oxygen or hydrogen.

In [87], Mochalski et al. showed the possibility to measure other gases transcutaneously or to target other biomarkers. Other studies have been conducted, by Arakawa, to measure ethanol [88] or, by Ohkuwa, to establish a link between hypoxia and NO concentration [89]. In studying exposure to pollutants, Sekine et al. [90] established a relation between toluene emanating from the skin and inhalation exposure.

## 6. Patents

GRANGEAT P., ACCENSI M., GHARBI S. and GRATEAU H. (2019), “Dispositif portable d’estimation de la pression partielle de gaz sanguin”, French patent demand number FR1906142, filed on 9 June 2019. “Portable device for estimating the partial pressure of blood gas”, international patent demand number PCT/EP2020/065523, filed on 4 June 2020, published on 17 December 2020, publication number WO 2020/249466 A1.

GRANGEAT P., STOCARD F. and JAILLET M.V. (2022), “Dispositif portable d’estimation d’une concentration de gaz dégagé par un milieu”, French patent demand number FR2208591, filed on 28 August 2022.

GRANGEAT P., COMSA M.P., KOENIG A., PHLYPO R. and A. KOENIG (2022), “Procédé d’estimation d’une concentration de gaz dégagé par un milieu”, French patent demand number FR2208595, filed on 28 August 2022.

## Figures and Tables

**Figure 1 sensors-23-06096-f001:**
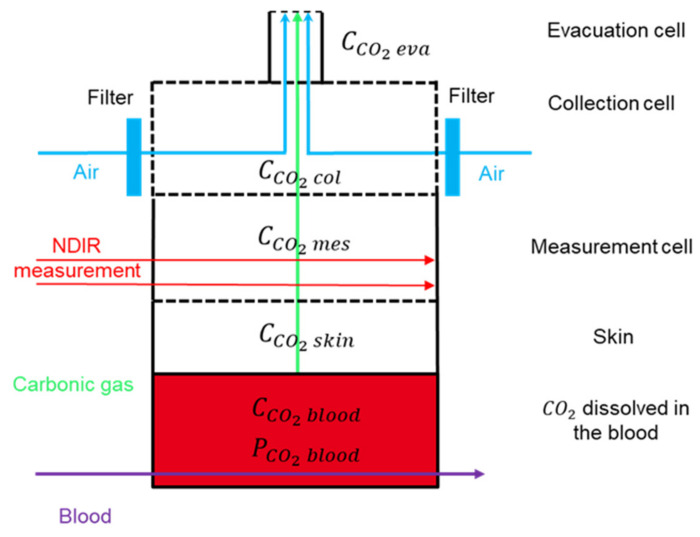
The operating mode of the capnometry wristband, CAPNO, relies on an NDIR optical measurement cell close to the surface of the skin and an air convection flow through the collection cell, which collects the $CO_{2}$ diffused from the blood through the skin and the measurement cell.

**Figure 2 sensors-23-06096-f002:**
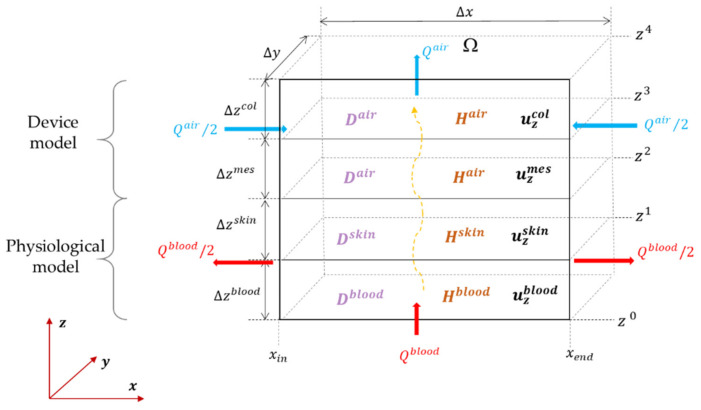
$CO_{2}$ desorption between blood and ambient air, and a system model based on the concatenation of our physiological and device models.

**Figure 3 sensors-23-06096-f003:**
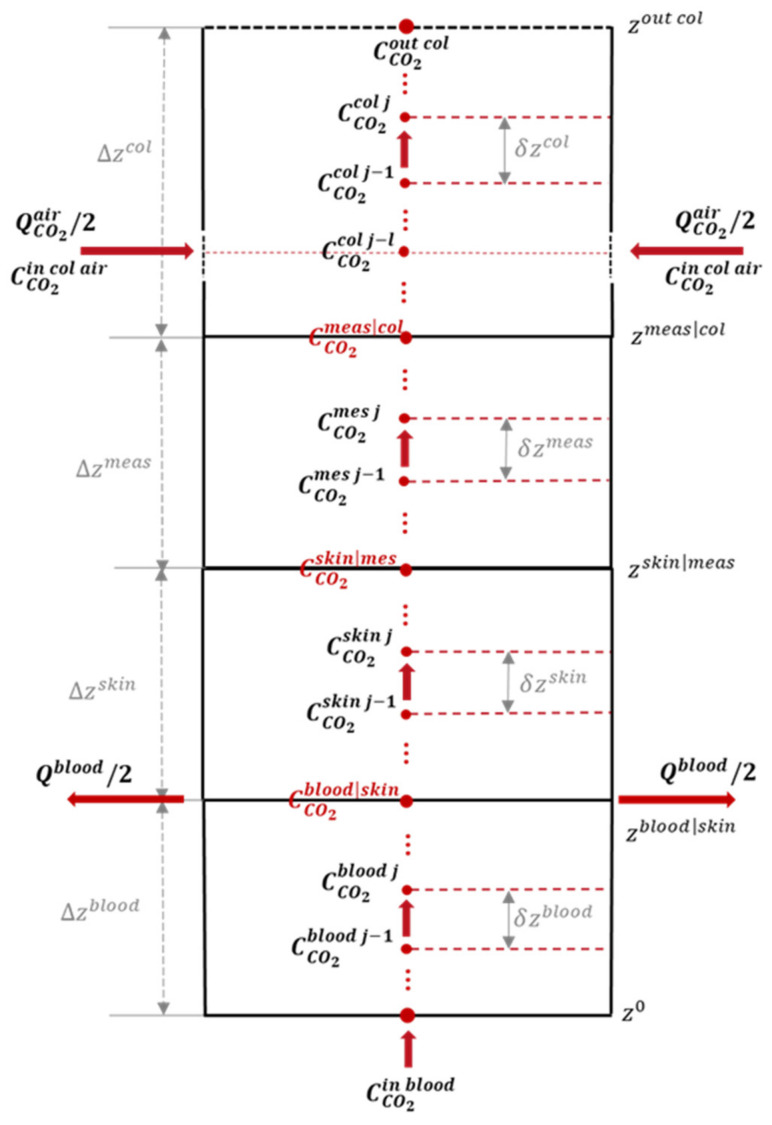
Geometrical model scheme and sample points considered for discretizing the continuous spatial space.

**Figure 4 sensors-23-06096-f004:**
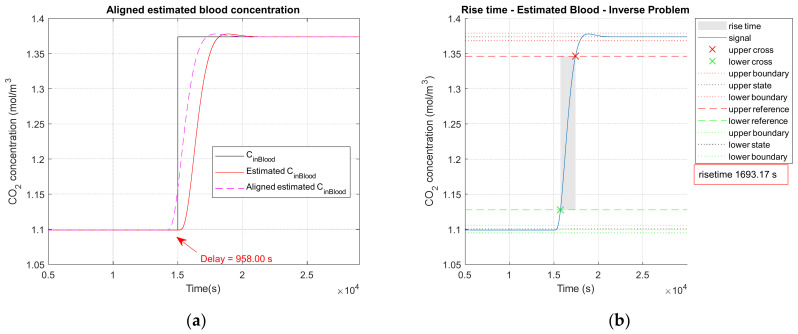
(**a**) Delay calculation; (**b**) rise time calculation. Noiseless observation vector of the Kalman inverse problem.

**Figure 5 sensors-23-06096-f005:**
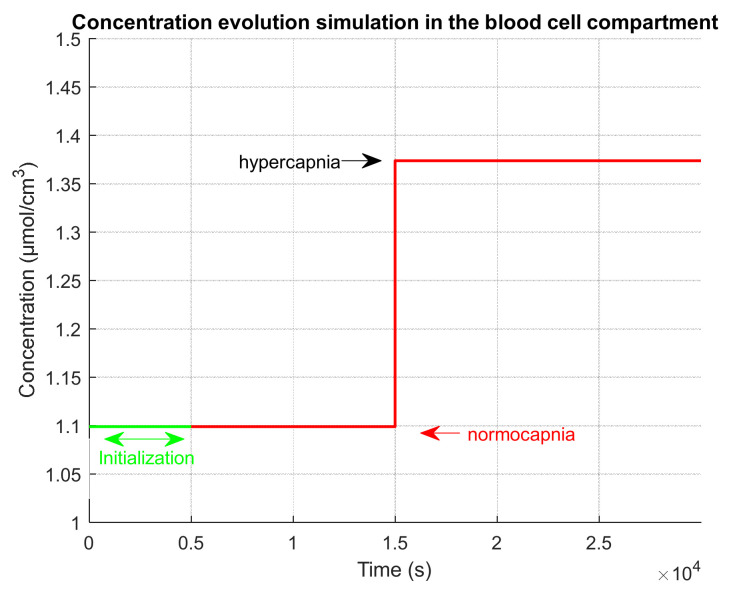
System input signal. The variations in blood concentration are considered as input signal for the direct approach. There are two levels of blood concentration: one corresponding to a normocapnia level and one corresponding to a hypercapnia level.

**Figure 6 sensors-23-06096-f006:**
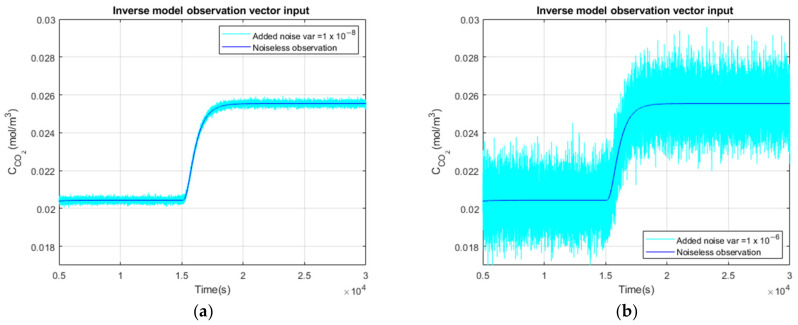
Input signal for the evaluation of the inverse transport model for the three cases proposed: when the input is a noiseless concentration signal (in dark blue); (**a**) when the input is characterized by a noise variance of $1\times 10^{-8}\;{\left(\mathrm{mol}/m^{3}\right)}^{2}$ (light blue); (**b**) when the input is characterized by a noise variance of $1\times 10^{-6}\;{\left(\mathrm{mol}/m^{3}\right)}^{2}$ (light blue).

**Figure 8 sensors-23-06096-f008:**
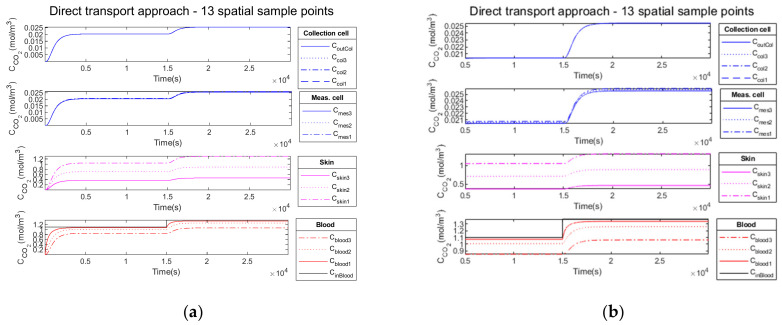
Synthetic data generated for 13 spatial sample points: the propagation of input concentration within the system through different media (**a**) with the initialization phase visible from 0 to normocapnia and (**b**) with the initialization phase not shown.

**Figure 9 sensors-23-06096-f009:**
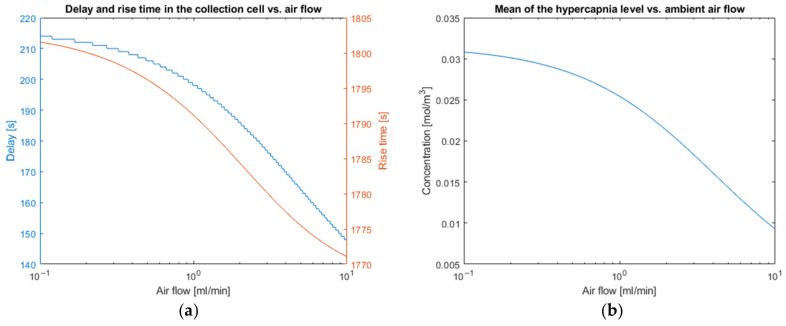
Device performance parameters: (**a**) the evolution of the time delay and rise time in the collection cell according to the flow of the ambient air entering the cell; (**b**) the variation of the mean of the hypercapnia measurement level according to the flow of the ambient air entering the cell.

**Figure 10 sensors-23-06096-f010:**
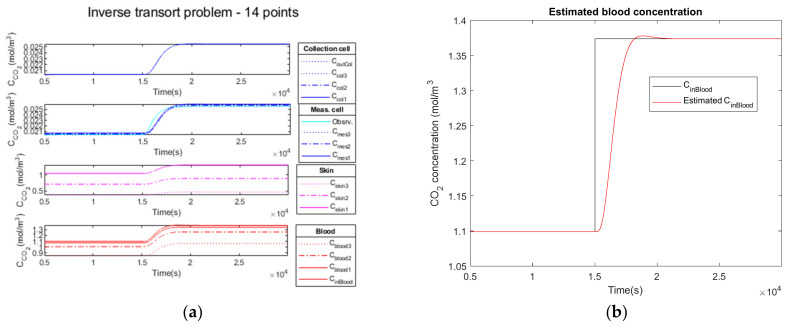
Noiseless observation of a step function as input. (**a**) Estimated concentrations using the spatial grid for the direct transport problem, including the variable $C_{CO_{2}}^{in\;blood}$ as a state variable. (**b**) Estimated (red curve) and real (black curve) blood concentration.

**Figure 11 sensors-23-06096-f011:**
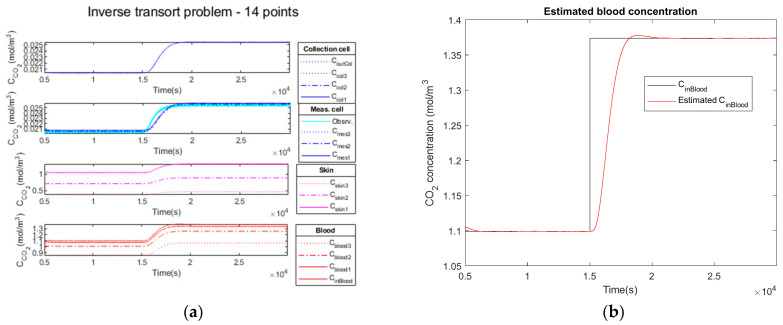
Low noise added to the observation of a step function as input. (**a**) Estimated concentrations using the spatial grid for the direct transport problem, including the variable $C_{CO_{2}}^{in\;blood}$ as a state variable. (**b**) Estimated (red curve) and real (black curve) blood concentration.

**Figure 12 sensors-23-06096-f012:**
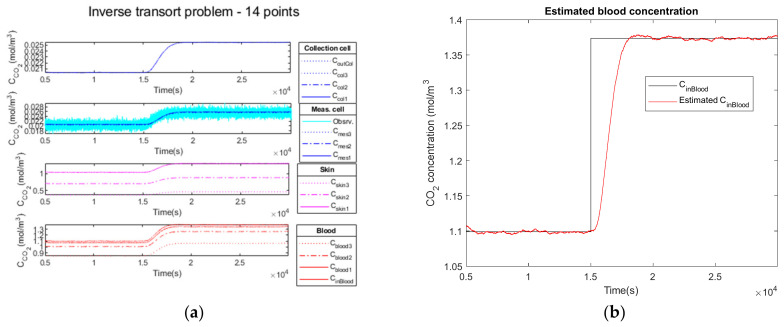
High noise added to the observation of a step function as input. (**a**) Estimated concentrations using the spatial grid for the direct transport problem, including the variable $C_{CO_{2}}^{in\;blood}$ as a state variable. (**b**) Estimated (red curve) and real (black curve) blood concentration.

**Figure 13 sensors-23-06096-f013:**
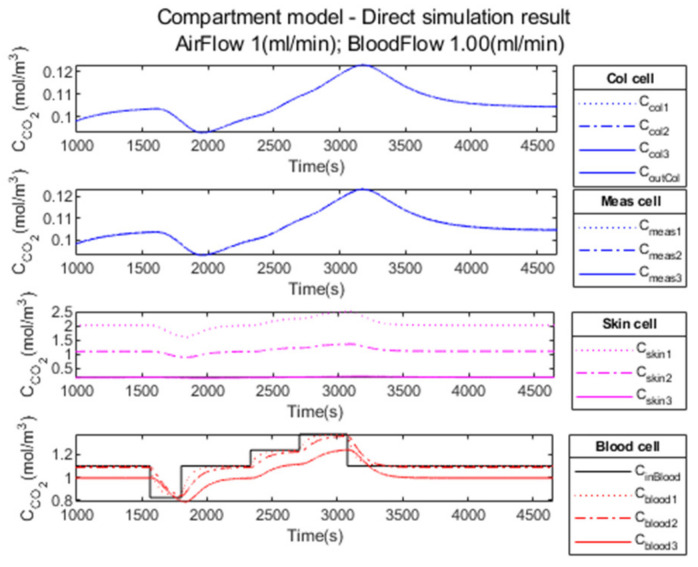
Result of the compartmental model simulation in the four compartments.

**Figure 14 sensors-23-06096-f014:**
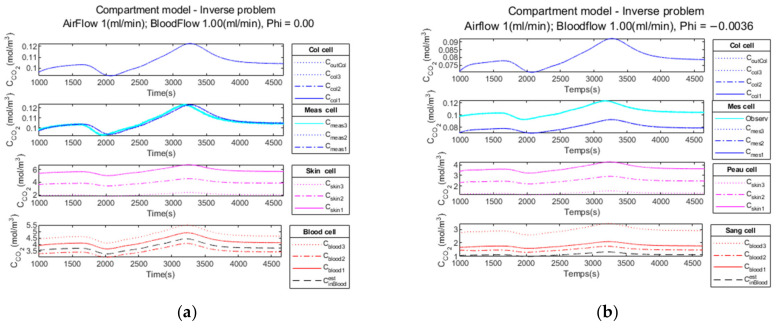
Result of the Kalman filter in the four compartments with two regularity parameters: (**a**) φ=0; (**b**) φ=−0.0036.

**Figure 15 sensors-23-06096-f015:**
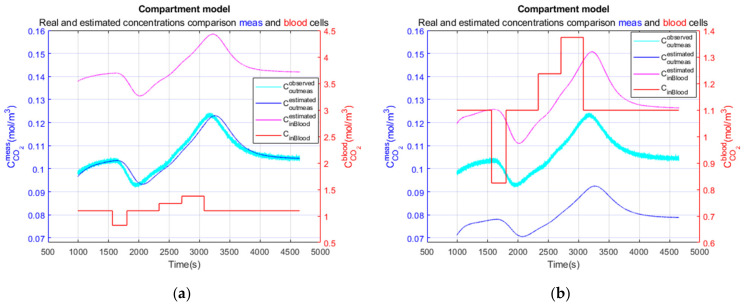
Comparison of simulated/estimated values in measurement and blood cell: (**a**) φ=0; (**b**) φ=−0.0036.

**Table 4 sensors-23-06096-t004:** Model parameters.

Parameter	Explanation	Set Member
N	Total number of points considered to discretize the continuous spatial space (state vector dimension)	ℕ
Ncomp	Total number of interior points considered in each compartment	ℕ
Δzcomp	Thickness value of each compartment	ℝ
c	State vector	ℝN×1
* **A** *	Transition matrix applied to state ck+1	ℝN×N
pinterface	Slack variable for compartment boundaries	
q	Command vector	ℝX×1
y	Data observation vector	ℝM×1
G	Command matrix	ℝN×X
h	Observation vector of observation response function	ℝN
w	Modelization noise vector	ℝN×1
v	Observation noise	ℝM×1
Q	Modelization noise covariance matrix	ℝN×N
R	Observation noise covariance matrix	ℝM×M
K	Kalman gain matrix	ℝN×M
F	Differential spatial operator	ℝN×N
φ	Signal regularity parameter	ℝ

**Table 5 sensors-23-06096-t005:** Kalman filter algorithm iteration notations.

Parameter	Explanation
$c_{k,k-1}$	state vector at time step k predicted at previous time step k−1
$c_{k,k}$	state vector at time step k computed at time step k
$c_{k,k+1}$	prediction of the state vector at the next time step, k+1
$P_{k,k-1}$	uncertainty covariance matrix at time step k predicted at previous time step k−1
$P_{k,k}$	uncertainty covariance matrix computed at the current state k
$P_{k,k+1}$	predicted uncertainty covariance matrix at the next time step, k+1
$K_{k}$	Kalman gain computed at the current state k
$y_{k}$	measurement at time step k

**Table 7 sensors-23-06096-t007:** Performance parameters for direct and inverse approach considering model noise variance of $10^{-8}\;{\left(\mathrm{mol}/m^{3}\right)}^{2}$ and observation noise variance of $10^{-6}\;{\left(\mathrm{mol}/m^{3}\right)}^{2},\;$ with regularity parameter φ=0, for three different levels of noise added to the observation. Kalman state vector is initialized to 0.

Parameters	Noiseless Observation	Low Noise Observation	High Noise Observation
$\mu_{hypercapnia}\left(\mathrm{mol}/m^{3}\right)$	1.3738	1.3738	1.3738
${\hat{\mu }}_{hypercapnia}\left(\mathrm{mol}/m^{3}\right)$	1.3736	1.3736	1.3733
$rmse\;\left(\mathrm{mol}/m^{3}\right)\;aligned$	0.0306	0.0306	0.0313
$perf_{rel}\;(\%)$	0.011	0.014	0.036
$rsb_{global}\;(\mathrm{dB})$	32.34	32.34	32.16
$t_{r}\;\left(s\right)$	1693	1694	1685
$t_{d-dir}\;\left(s\right)$	618	618	618
$t_{d-inv}\;\left(s\right)$	0	0	0
$t_{d-global}\;\left(s\right)$	958	959	931

**Table 8 sensors-23-06096-t008:** Performance parameters per capnia phase, $CO_{2}$ mean value, relative performance and rise time for direct and inverse approach considering model noise variance of $10^{-8}\;{\left(\mathrm{mol}/m^{3}\right)}^{2}$ and observation noise variance of $10^{-6}\;{\left(\mathrm{mol}/m^{3}\right)}^{2}$ for two different values of parameter.

Parameters	Phase 1 Normocapnia		Phase 2 Hypocapnia		Phase 3 Normocapnia		Phase 4 Hypercapnia 1		Phase 5 Hypercapnia 2		Phase 6 Normocapnia	
(φ)	0	−0.0036	0	−0.0036	0	−0.0036	0	−0.0036	0	−0.0036	0	−0.0036
$\mu_{}\left(\mathrm{mol}/m^{3}\right)$	1.099	1.099	0.825	0.825	1.099	1.099	1.236	1.236	1.374	1.374	1.099	1.099
${\hat{\mu }}_{}\left(\mathrm{mol}/m^{3}\right)$	3.668	1.094	3.416	1.019	3.583	1.069	4.003	1.19	4.42	1.317	3.841	1.145
$perf_{rel}$ (%)	−234	0.45	−313	−23.49	−226	2.74	−224	3.71	−222	4.16	−249	−4.17
$t_{r}\;\left(s\right)$			214.36	215.70	251.90	264.16	264.37	254.01	225.39	225.61	519.61	529.53

**Table 9 sensors-23-06096-t009:** Global performance parameters: rmse of align signals, global RSB and delays for direct and inverse approach. We consider model noise variance of $10^{-8}\;{\left(\mathrm{mol}/m^{3}\right)}^{2}$ and observation noise variance of $10^{-6}\;{\left(mol/m^{3}\right)}^{2}$ for two different values of parameter φ.

Parameters	Values (φ=0)	Values (φ=−0.0036)
$rmse_{align}\left(\mathrm{mol}/m^{3}\right)$	2.68	0.065
$rsb_{global}$ (dB)	−7.48	24.82
$t_{d-dir}\;\left(s\right)$	330.03	330.03
$t_{d-inv}\;\left(s\right)$	32.09	41.2
$t_{d-global}\;\left(s\right)$	369.65	369.05

## Data Availability

Not applicable.

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
