# Peer review of "Dynamic Modeling of Carbon Dioxide Transport through the Skin Using a Capnometry Wristband"

_sensors, 2023, doi:10.3390/s23136096_

Round 1

Reviewer 1 Report

 The auther considered a new architecture where a NDIR optical measurement is located close to the surfaceof the skin and is combined with an open chamber principle with a continuous circulation of an air flow in the collection cell. This dynamic model  is a key contribution for the development of an autonomous wearable device. The work is well, but it still needs some minor revison. The  introduction needs to be revised and some references suposed to be included, such as Nature 576, 220-221 (2019), Sensors and Acuators : B Chemical (390)2023,133894 et al.

Reviewer 2 Report

I have reviewed the manuscript carefully entitle “Dynamic modeling of carbon dioxide transport through the skin on a capnometry wristband”. The topic is useful and interesting but manuscript needs some modification before acceptance. My concerns are given below.

·      Authors need to define abbreviation when it comes first time.   

·      The language of manuscript should be standardized and authors need to avoid “ we…” Sentences should be change in generic and scientific language.

·      Abstract is too general and needs revision with pacific numeric information.

·      Authos performed study about Co2, What about the other gasis? Need description.

·      How Kalman filter different from other filters need justification why they have used in their study.

·      Authors need to add citation in mathematical equations which are generic.

·      Skin exhibit hypo elastic properties, Authors have ignored flow behavior of Co2 with skin properties.

·      Authors need to add some more detail of behavior of model with temperature variation

·      Manuscript is lack of focus discussion and authors need to add detailed discussion.

·       Bench mark table should be added by giving the compression of current study with literature by giving its pros and cons.

·       Conclusion is also generic and need to add concluding remarks with significance point by point instead of summery

Need correction and revision of language

Reviewer 3 Report

Thank you for the opportunity to review this paper.

This study bring new to  consider an open chamber principle with a continuous air  flow.

I do not have the necessary knowledge to review the proposed model. The development of a wearable device for continuous monitoring of blood carbon dioxide content is a topic of great interest for home care.

Reviewer 4 Report

In the manuscript “Dynamic modeling of carbon dioxide transport through the skin on a capnometry wristband”, Grangeat et al proposed a model for the temporal dynamics of the carbon dioxide exchange between blood and the gas channel within the device. Convection—diffusion equations were used to model carbon dioxide transportation. Four compartments, namely blood, skin, measurement cell and collection cell were considered. Several parameters for the direct model, the inverse model, and the complete model on a realistic clinical test were studied. The development of a wearable capnometry device is a requirement for continuous monitoring of COPD patient. The development of such a capnometry device is a challenging topic because the transcutaneous carbon dioxide flow is very low. Thus, the authors proposed a dynamic model of carbon dioxide transport through the skin on a capnometry wristband. Such a dynamic model is a requirement to design an accurate instrumentation, compute simulated data, and develop a model-based signal processing. Accordingly, such a model is a key contribution for the development of an autonomous wearable device. Many patents involved in this dynamic modeling of carbon dioxide transport were also applied. This report is very interesting and useful for further medical applications. This reviewer recommends publication.

Reviewer 5 Report

The article focuses on developing a capnometry wristband for homecare patient monitoring. The authors propose a new architecture involving a near-infrared (NDIR) optical measurement device close to the skin surface. This device is combined with an open chamber principle that allows continuous airflow circulation in the collection cell. The authors present a model to describe the temporal dynamics of carbon dioxide exchange between the blood and the gas channel within the wristband.

The authors employ convection-diffusion equations to model carbon dioxide transportation and consider four compartments: blood, skin, measurement cells, and collection cells. They introduce state-space equations and a corresponding transition matrix using a Markovian model. To account for the blood's resistance to change, they incorporate a first-order autoregressive carbon dioxide concentration input model in the blood compartment. The authors propose using a Kalman filter for real-time estimation of carbon dioxide concentration in the blood vessels, allowing monitoring of carbon dioxide blood pressure.

The article discusses four performance factors related to dynamic carbon dioxide blood concentration quantification. A simulation is conducted using data from a previous clinical study to demonstrate the feasibility of the proposed technology concept.

The topic's significance lies in developing a capnometry wristband that enables non-invasive monitoring of carbon dioxide blood pressure in homecare settings. This approach can enhance patient comfort, convenience, and overall monitoring accuracy.

The method used in the study combines mathematical modeling, specifically convection-diffusion equations and a Markovian model, with the implementation of a Kalman filter for estimation. The authors describe the model and its associated equations, which provide a theoretical foundation for the proposed wristband design. However, it is essential to note that the article does not provide details on the wristband's specific implementation or experimental setup. The paper lacks recent citations related to the problem and processing of such temporal data. The article below and other articles may be very helpful: https://doi.org/10.3390/bioengineering10020249

The results presented in the article are based on a simulation using data from a previous clinical study. While the feasibility of the technology concept is demonstrated, it is crucial to emphasize that real-world validation through practical experiments and clinical trials is necessary to confirm the accuracy and reliability of the proposed capnometry wristband.

The conclusions drawn from the study highlight the potential of the proposed architecture and modeling approach for developing a capnometry wristband. However, further research is needed to validate the performance and efficacy of the wristband in real-world scenarios and patient populations.

Overall, the article introduces the concept of a capnometry wristband for homecare patient monitoring and presents a mathematical modeling framework for its development. However, additional experimental studies and clinical validations are required to fully evaluate this technology's significance and potential impact.

The language should be revised by a native speaker or certified proofreading should be submitted. Bullets and lack of linkages among sentences should be avoided.

Round 2

Reviewer 5 Report

No Further comments.